# Personalized Decision Modeling: Utility Optimization or Textualized-Symbolic Reasoning

**Yibo Zhao**
Department of Civil and Systems Engineering
Johns Hopkins University

**Yang Zhao**
Department of Civil and Systems Engineering
Johns Hopkins University

**Hongru Du** [*][†]
Department of Systems and Information Engineering
University of Virginia
hongrudu@virginia.edu

**Hao Frank Yang** [†]
Department of Civil and Systems Engineering
Johns Hopkins Data Science and AI Institute
Johns Hopkins University
haofrankyang@jhu.edu

## Abstract

Decision-making models for individuals, particularly in high-stakes scenarios like vaccine uptake, often diverge from population optimal predictions. This gap arises from the uniqueness of the individual decision-making process, shaped by numerical attributes (e.g., cost, time) and linguistic influences (e.g., personal preferences and constraints). Developing upon Utility Theory and leveraging the textual-reasoning capabilities of Large Language Models (LLMs), this paper proposes an Adaptive Textual-symbolic Human-centric Reasoning framework (**ATHENA**) to address the optimal information integration. ATHENA uniquely integrates two stages: First, it discovers robust, group-level symbolic utility functions via LLM-augmented symbolic discovery; Second, it implements individual-level semantic adaptation, creating personalized semantic templates guided by the optimal utility to model personalized choices. Validated on real-world travel mode and vaccine choice tasks, ATHENA consistently outperforms utility-based, machine learning, and other LLM-based models, lifting F1 score by at least 6.5% over the strongest cutting-edge models. Further, ablation studies confirm that both stages of ATHENA are critical and complementary, as removing either clearly degrades overall predictive performance. By organically integrating symbolic utility modeling and semantic adaptation, ATHENA provides a new scheme for modeling human-centric decisions. The project page can be found at https://yibozh.github.io/Athena.

## 1 Introduction

Consider the widely debated *vaccine dilemma* [1], from a population-level perspective aimed at optimizing collective well-being (e.g., achieving herd immunity at minimal societal cost), models would invariably predict near-universal vaccine adoption. However, this population optimum consistently

---

[*]This work was completed while Hongru Du was at Johns Hopkins University.
[†]Correspondence to: Hongru Du and Hao Frank Yang.

39th Conference on Neural Information Processing Systems (NeurIPS 2025).

fails to predict actual individual behavior. The reality is a broad spectrum of personal choices, because each individual undertakes their own unique *cognitive calculus*: balancing a vaccine's perceived efficacy and protection against its perceived risks [2], all filtered through their personal beliefs, constraints, and even considerations of potential "free-riding" on the immunity of others [3]. This divergence between the theoretical population optimum and observed individual actions highlights a critical limitation: **models designed to maximize collective outcomes do not adequately explain or predict individuals' choices.** Instead, individual choices are profoundly shaped by who we are, when the decision is made, and the unique constraints we face. This internal *'cognitive calculus'*, unique to each individual and situation, presents a profound challenge for human decision modeling.

For decades, researchers have attempted to model this *'cognitive calculus'* with Utility Theory [4–6], which assumes individuals select options that maximize expected gain. Operationally, this involves defining a parametric utility function, denoted as $f : \mathcal{X} \to \mathbb{R}$, that maps a vector of structured attributes $\mathcal{X}$ (e.g., monetary cost and time) for each option to a scalar utility score. Such pre-defined and explicit specifications of $f$ are the basis for classic discrete choice models [7], which have been widely adopted across economics [8–10], transportation [7, 11–13], and public policy [14–16]. In these models, the utility scores derived from $f$ for each available option are used to probabilistically determine the likelihood of an individual selecting a particular option. However, even these utility-based models encounter fundamental barriers when attempting to capture the full depth of human decision-making. Real-world human decisions, as seen in the *vaccine dilemma*, frequently deviate from these mathematical formulations. Individuals exhibit behaviors that appear inconsistent or irrational [17, 18], yet these are often driven by subjective feelings and personal experiences. Such deviations reflect that traditional models, with their reliance on pre-specified functions, struggle to capture personalized decisions [19]. The clue for deciphering this deviation is covered within individual attributes, some of which are structured and quantifiable, while others are unstructured and semantic (e.g., individual preference and constraints).

Addressing these unstructured and semantic dimensions, which are pivotal for capturing personalized decision mechanisms, calls for new modeling paradigms. LLMs with their strong textual-reasoning capability offer a clear advance [20], providing new mechanisms for identifying the utility function $f$ and for integrating semantic individual context directly into the decision modeling process. Specifically, LLMs enhance our ability to model human choice by: a) Guiding the discovery of more accurate and robust parametric forms for $f$. Through LLM-augmented symbolic regression [21–23], it becomes feasible to identify data-driven mathematical structures that capture underlying group-level choice patterns more effectively than pre-specified forms. b) Enabling the infusion of individual-level textual information into the human decision modeling [24–26]. By encoding personal preferences, constraints, and narratives, LLMs allow each decision to reflect the nuanced motivations and situational factors that traditional numeric features alone cannot convey.

This paper introduces an **Adaptive Textual-symbolic Human-centric Reasoning framework (ATHENA)**. ATHENA achieves personalized decision modeling by uniquely integrating two sequentially structured steps: First, at the group level, it focuses on discovering robust, symbolic utility functions. Second, it implements individual-level, LLM-powered semantic adaptation guided by optimal utility functions discovered in previous steps. The outcome is a customized semantic template for each person, specifically designed to empower an LLM to model their choices by incorporating their unique preferences and constraints. We empirically validate ATHENA on two real-world human decision-making tasks: travel mode choice and vaccine uptake decisions. The model consistently outperforms traditional utility-based, machine learning, and LLM-based models, with at least $6.5\%$ improvement in F1 score. Further ablation experiments reveal that removing either the group-level symbolic utility search or the individual semantic adapter lowers performance by at least $18\%$, underscoring the merit of the full ATHENA framework.

## 2 Related Work

**Utility-based Decision-Making Models.** Initial explorations into this complex domain were predominantly by utility-based models [27–30]. These methods aim to capture human behavior within explicit mathematical functions, formulated from empirical data. Established methodologies such as Discrete Choice Models (DCMs) have been widely used, attributable to their interpretability and robust statistical underpinnings [31, 10, 32–37]. While offering tractability, they may also limit the ability to fully capture complex non-linear patterns and diverse preferences in modern high-dimensional data.

**Machine Learning–Driven Decision-Making Models.** ML-driven decision-making models aim to directly learn from rich, diverse features. Tree-based ensemble methods, including Random Forests [38], Gradient-Boosting Trees [39], XGBoost [40, 41], and LightGBM [42], alongside neural network architectures [43–45], exhibited a notable proficiency in fitting complex, non-additive interaction effects. These models effectively integrated large-scale data, but their decision-making processes often lacked transparency despite strong predictive performance. Efforts to enhance transparency via post-hoc explanation frameworks, for instance, SHAP [46–48] and Integrated Gradients [49, 50], have provided some insights for human behavior. A persistent challenge is these models' vulnerability to distribution shifts, lack of transparency, and limited ability to provide symbolic, interpretable insights needed for personalized utility reasoning.

**Symbolic Regression with LLMs.** Classical symbolic regression (SR) typically uses genetic programming to evolve populations of candidate equations via stochastic mutation and crossover [51–53]. The goal is to find formulas that balance simplicity, generalizability, and human interpretability [54–56]. The recent rise of LLMs has revitalized symbolic regression, enabling new possibilities in scientific discovery when combined with advanced evolutionary algorithms [57]. For example, LLM-SR integrates LLM with evolutionary symbolic regression by treating equations as executable programs. It leverages LLMs' scientific prior knowledge and code generation abilities to iteratively generate, refine, and optimize equation skeletons [22]. LASR integrated LLM-driven abstract textual concepts within evolutionary frameworks, achieving notable performance enhancements on benchmarks, such as the Feynman equation set [58]. The DiSciPLE framework extended these contributions by emphasizing the interpretability and reliability of generated scientific hypotheses, incorporating critical evaluation and simplification to ensure hypotheses are both scientifically rigorous and computationally efficient [59].

**LLM-based Decision-Making Models.** The advent of LLMs has offered a new opportunity, establishing these models as human-like reasoning engines [60–65]. Techniques such as instruction tuning [66–69], chain-of-thought reasoning [70–74] are elevating LLMs move beyond basic text generation to handle more complex tasks involving step-by-step reasoning and symbolic or numerical problem-solving [75–81]. Within the specific domain of personalized decision making, preliminary findings suggest that zero-shot and few-shot prompting strategies can enhance the behavioral alignment of LLMs [82, 83]. Because a model's knowledge is inherited from generic pre-training priors, its reasoning defaults to universally salient factors – e.g., cost and time in travel mode choice – while overlooking personal preferences such as rail-pass loyalty or transfer aversion, thereby introducing systematic bias [84]. Techniques like persona loading partially mitigate this gap by conditioning responses on inferred preference structures [82, 85]. Beyond basic prompting, decision-centric systems add explicit structure to improve reliability and transparency: *DeLLMa* enumerates plausible states, elicits utilities via pairwise comparisons, and maximizes expected utility; *STRUX* distills inputs into fact tables with self-reflective evidence; *OptiGuide* compiles natural-language "what-if" queries into optimization code and invokes solvers; *Agent-Driver* coordinates tool calls, commonsense/experience memory, and chain-of-thought planning; *Personalized Oncology* evaluations show chat-LLMs still trail experts, motivating structured, evidence-grounded pipelines [86–90]. Nevertheless, many deployments still treat LLMs as opaque scoring mechanisms, falling short of fully recovering explicit, personalized utility logic.

# 3 Methods

We consider a classic discrete choice problem, where an individual $i$ faces a finite set of choices $\mathcal{J} = \{1, 2, \ldots, J\}$, where $J = |\mathcal{J}|$. The decision-making process assumes individuals select the option $j$ that maximizes their utility. Each individual's observed choice behavior is represented by a dataset $\mathcal{D} = \{(X_i, y_i)\}_{i=1}^{N}$. For each observation $i$, $X_i = \{x_{ij}\}_{j=1}^{J}$ represents the set of feature vectors, where $x_{ij} \in \mathbb{R}^K$ captures the features for choice $j$ and $y_i \in \mathcal{J}$ denotes the observed choice.

The standard approach to modeling discrete choices is the Random Utility Maximization (RUM) framework [91]. It assumes the latent utility for each alternative $j$ is described by $U_{ij} = f(X_i, j) + \epsilon_{ij}$, where $f(X_i, j)$ is systematic component of utility and $\epsilon_{ij}$ is the random error. Assuming $\epsilon_{ij}$ are independently and identically drawn (i.i.d.) and follow a Type I Extreme Value distribution [92], the probability of individual $i$ choosing alternative $j$ is given by:

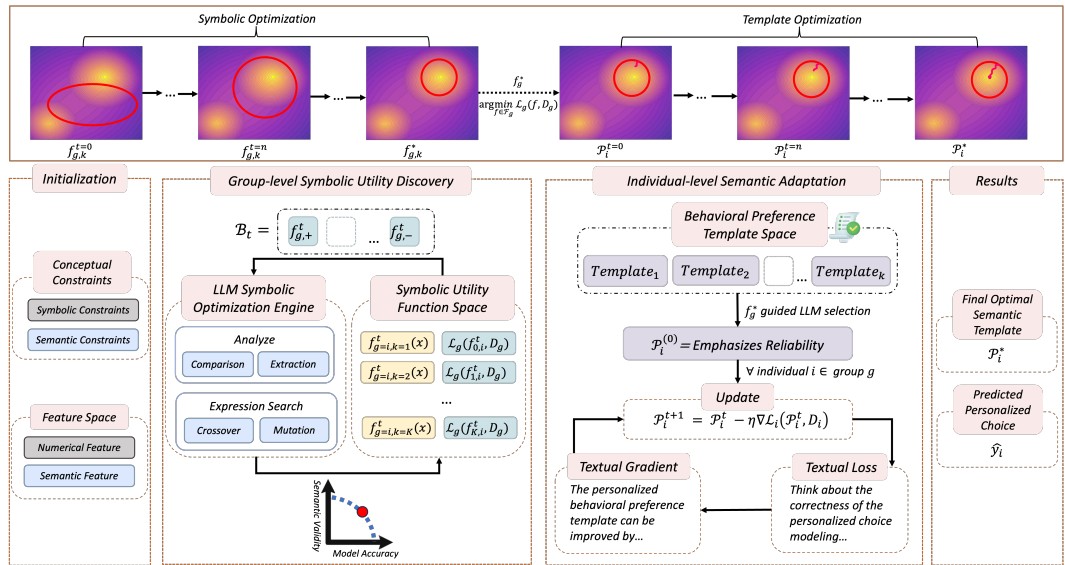

Figure 1: **Overview of the proposed ATHENA framework.** *Group-level symbolic utility discovery*: Symbolic & semantic constraints library feed an LLM-driven symbolic-optimization engine that iteratively proposes candidate utility functions, scores them with loss $\mathcal{L}_g$, and prunes the search via analysis, crossover, and mutation. Red rings in the contour maps illustrate how the feasible solution space shrinks across iterations until the optimal formula $f_g^*$ is selected. *Individual-level semantic adaptation*: The optimal group utility $f_g^*$ seeds a personalized template space. For each individual $i$, TextGrad computes textual gradients of an individual loss and updates the template $\mathcal{P}_i^t$ into a more personalized decision rule $\mathcal{P}_i^{t+1}$. Finally, the optimal $\mathcal{P}_i^*$ is used to predict personal decisions.

$$P(y_i = j \mid X_i) = \frac{e^{f(X_i, j; \theta)}}{\sum_{k \in \mathcal{J}} e^{f(X_i, k; \theta)}}. \tag{1}$$

A central challenge lies in specifying the systematic utility component $f$. Traditional applications of RUM often rely on pre-specified functional forms for $f$ using domain expertise and observed data [93]. This approach may result in a suboptimal representation of the true decision mechanism, while also neglecting individual heterogeneity in choices. Furthermore, traditional decision-making models are not designed to incorporate non-structured semantic information. To address these limitations, we introduce ATHENA for personalized decision modeling designed to identify suitable utility function representations while simultaneously capturing individual-specific preferences. As shown in Fig. 1, ATHENA structures the decision modeling process into two sequential stages:

1. **Group-Level Symbolic Utility Discovery:** This initial stage focuses on identifying optimal symbolic utility components that capture common decision patterns within distinct demographic groups. The discovery is achieved through a feedback-informed symbolic discovery process powered by LLMs.

2. **Individual-Level Semantic Adaptation:** Then, the optimal group-level utility functions serve as guidance for the LLM-driven optimization of personalized semantic templates. This adaptation process is designed to incorporate individual-specific preferences and constraints, leveraging the rich semantic reasoning capabilities of LLMs.

## 3.1 Group-Level Symbolic Utility Discovery

The first stage aims to discover an optimal parametric utility function, denoted as $f_g^*$, for each demographic group $g \in \mathcal{G}$. This function $f_g^*$ should be constructible from symbolic building blocks and optimally explain the group's choice behavior. Following the choice probability defined earlier

Eq. (1), the objective is to find the optimal $f_g^*$ and its associated parameters $\theta_g^*$ such that:

$$(\theta_g^*, f_g^*) = \underset{f,\theta}{arg\,min} \, \mathcal{L}_g(f(X_i, y_i; \theta); \mathcal{D}_g) = - \sum_{(X_i, y_i) \in \mathcal{D}_g} \log \left( \frac{e^{f(X_i, y_i; \theta)}}{\sum_{k \in \mathcal{J}} e^{f(X_i, k; \theta)}} \right). \quad (2)$$

To automate the symbolic utility discovery of $f_g^*$, we design an iterative, feedback-informed generation process powered by LLMs. To effectively guide the automated discovery of utility functions, we constructed two foundational libraries: a domain knowledge concept library ($\mathcal{C}$) and a symbolic library ($\mathcal{S}$). The library $\mathcal{C}$, developed based on input from domain experts, covers high-level conceptual knowledge about domain-specific human behavior. The library $\mathcal{S}$ provides the fundamental syntactic building blocks needed for constructing all candidate utility expressions.

Inspired by evolutionary algorithms [94], the core discovery process proceeds iteratively. In each iteration $t$ for each demographic group $g$, the LLM samples a set of $K$ candidate symbolic utility functions, $\{f_{g,k}^t\}_{k=1}^K$. This sampling is performed from the LLM's learned distribution $\phi$ [20, 95], conditioned on the group profile $g$, the domain concept $\mathcal{C}$, the available symbolic building block $\mathcal{S}$, and a feedback $\mathcal{B}^{t-1}$ from preceding iteration:

$$\{f_{g,k}^t\}_{k=1}^K \sim \phi(\cdot | g, \mathcal{C}, \mathcal{S}, \mathcal{B}^{t-1}) \quad (3)$$

The feedback $\mathcal{B}$ is essential in refining the LLM's sampling strategy. Specifically, $\mathcal{B}^t$ is constructed at the end of each iteration t and comprises the best-performing and worst-performing candidate functions from that iteration:

$$\mathcal{B}^t = \{f_{g,+}^t, f_{g,-}^t\}, \quad (4)$$

where $f_{g,+}^t = \underset{k \in K}{arg\,min} \, \mathcal{L}_g(f_{g,k}^t, \mathcal{D}_g)$ and $f_{g,-}^t = \underset{k \in K}{arg\,max} \, \mathcal{L}_g(f_{g,k}^t, \mathcal{D}_g)$. Here $\mathcal{L}_g$ is the group-level loss function, with a similar format as Eq. (2). This feedback $\mathcal{B}^t$ is used to refine the LLM's sampling distribution $\phi$ through stochastic *mutation* or *crossover* [51–53], pushing the generation towards more promising types of functions. The iterative discovery process for group $g$ is considered to have converged at iteration $T$ if the absolute difference in the loss of the best-performing function from the current iteration $t$ and that of the previous iteration $t-1$ falls below a predefined threshold $\delta$:

$$|\mathcal{L}_g(f_{g,+}^t, \mathcal{D}_g) - \mathcal{L}_g(f_{g,+}^{t-1}, \mathcal{D}_g)| < \delta. \quad (5)$$

Upon convergence at iteration at $T$, the optimal group-level symbolic utility function ($f_g^*$) is determined as the function that achieved the minimum loss across all generated candidate functions throughout the entire iterative process:

$$f_g^* = \arg \min_{f \in \mathcal{F}_g} \mathcal{L}_g(f, D_g), \quad (6)$$

where $\mathcal{F}_g = \bigcup_{t=1}^T \{f_{g,k}^t\}_{k=1}^K$ and $T$ is the iteration at which convergence occurred. This discovered function $f_g^*$, along with its fitted parameters $\theta_g^*$, serves as the learned representation of the systematic utility for group $g$.

### 3.2 Individual-Level Semantic Adaptation

Following the determination of the group-level optimal symbolic utility functions $f_g^*$, the framework transitions to the second stage, leveraging an LLM conditioned on $f_g^*$ to model individual choice behavior more accurately. While $f_g^*$ captures the central tendencies of utility for group $g$, significant intra-group heterogeneity often persists. To account for this, we introduce an individual-level adaptation stage to personalize the utility representation by generating and refining an individual-specific semantic template.

For each individual $i \in g$, the initial semantic template, denoted as $\mathcal{P}_i^0$ is generated by the LLM ($\phi$). The generation of the initial semantic template is represented as a sampling process from the LLM's distribution: $\mathcal{P}_i^0 \sim \phi(\cdot | f_g^*, i, \mathcal{C})$. In this formulation, $\phi$ conditions on the optimal group-level symbolic function $f_g^*$, the specific individual context $i$, and the high-level domain concepts from $\mathcal{C}$ to generate $\mathcal{P}_i^0$. This initial template $\mathcal{P}i^0$ is a semantic representation that is designed to be adaptable in

subsequent optimization steps. Then, the semantic template $\mathcal{P}_i^0$ undergoes an iterative refinement process for each individual $i$. This optimization is driven by TextGrad [96], which optimizes the template based on the individual's specific data $\mathcal{D}_i = (X_i, y_i)$. The update rule is given by:

$$\mathcal{P}_i^{t+1} \leftarrow \mathcal{P}_i^t - \eta \nabla \mathcal{L}_i(\mathcal{P}_i^t, D_i). \tag{7}$$

The term $\nabla \mathcal{L}_i(\mathcal{P}_i^t, D_i)$ represents the "textual gradient" of the loss function with respect to the semantic template $\mathcal{P}_i^t$. Since $\mathcal{P}_i^t$ is the textual template, this gradient is not a vector of partial derivatives in the mathematical sense. Instead, it indicates the direction and nature of textual modifications to $\mathcal{P}_i^t$ that would lead to the most improvement in loss. This iterative refinement process continues until a maximum number of iterations $T'$ is reached. Then the final optimal semantic template for individual $i$, denoted as $\mathcal{P}_i^*$, is determined. The predicted personalized choice $\hat{y}_i$ is then represented as sampling from the LLM's output distribution:

$$\hat{y}_i \sim \phi(\ \underbrace{\mathcal{P}_i^*, X_i}_{\text{Semantic Adaptation}}\ |\ \overbrace{f_g^*(X_i; \theta_g^*)}^{\text{Symbolic Utility Discovery}}\ ) \tag{8}$$

The overall procedure of ATHENA is summarized in Algorithm 1.

## 4 Experiments

This section empirically validates the value of ATHENA, demonstrating its overall effectiveness in personalized decision-making and its robust capability to apply across diverse application domains. We break down our experimental findings to specifically showcase the distinct value added by each core component of the ATHENA framework: 1) group-level symbolic utility discovery and 2) personalized semantic template adaptation. Fig. 2 illustrates the full pipeline using the travel-mode choice as an example.

### 4.1 Experimental Setup

**Datasets.** To test ATHENA's ability to generalize across different domains and to adapt to individual preferences,

---

**Algorithm 1** ATHENA Optimization Flow

**Require:** Demographic group $g$, dataset $\mathcal{D}_g$, domain concept $\mathcal{C}$, symbolic building block $\mathcal{S}$
1: Initialize $\mathcal{B}_0 \leftarrow$ None
   *// Stage 1: Group-Level Symbolic Utility Discovery*
2: **for** $t = 1$ to $T$ **do**
3:     Sample symbolic utility functions $\{f_{g,k}^t\}_{k=1}^K \sim \phi(\cdot \mid g, \mathcal{C}, \mathcal{S}, \mathcal{B}^{t-1})$
4:     Update $\mathcal{B}^t \leftarrow \{f_{g,+}^t, f_{g,-}^t\}$ using Eq. (4)
5:     Select best function $f_g^* \leftarrow \arg\min_{f \in \mathcal{F}_g} \mathcal{L}_g(f, \mathcal{D}_g)$
6:     **if** stopping condition in Eq. (5) is met **then**
7:         **break**
8:     **end if**
9: **end for**
   *// Stage 2: Individual-Level Semantic Adaptation*
10: **for** each individual $i \in g$ **do**
11:     Initialize semantic template $\mathcal{P}_i^0 \sim \phi(\cdot \mid f_g^*, i, \mathcal{C})$
12:     **for** $t = 1$ to $T'$ **do**
13:         Update $\mathcal{P}_i^{t+1} \leftarrow \mathcal{P}_i^t - \eta \nabla \mathcal{L}_i(\mathcal{P}_i^t, \mathcal{D}_i)$ using Eq. (7)
14:     **end for**
15: **end for**
16: **return** $\{\mathcal{P}_i^*\}_{i \in g}$, predict decisions using Eq. (8).

---

we selected two real-world tasks that reflect fundamentally different personalized decision scenarios: daily transportation choices and public health decisions. **(1) Swissmetro Transportation Choice (*Swissmetro*)**: is a widely used benchmark in travel mode choice modeling [97–101]. Each record details a trip between major Swiss cities and includes both traveler characteristics (e.g., income, age) and alternative-specific attributes (e.g., travel time, cost). The dataset has a potential choice set of three transportation modes: *Train*, *Car*, and *Metro*. **(2) COVID-19 Vaccination Choice (*Vaccine*)**: This dataset is derived from a large-scale international survey, conducted across multiple countries [102]. The survey was designed to understand factors influencing COVID-19 vaccine uptake and attitudes. For each participant, it captures demographics, prior beliefs about the vaccine, and their self-reported vaccination status. The modeled choices based on this information include: *Unvaccinated*, *Vaccinated initial doses, no booster*, and *Vaccinated initial doses plus booster*.

**Experiment Configurations.** To maintain a reasonable budget for the template-adaptation stage, we restricted the experimental sample to a representative subset of each dataset. Specifically, we used: *(1) Swissmetro*: 500 travelers, two trip records per person; *(2) Vaccine*: 300 respondents, one survey record per person. Within each dataset, we first identified key demographic dimensions (gender,

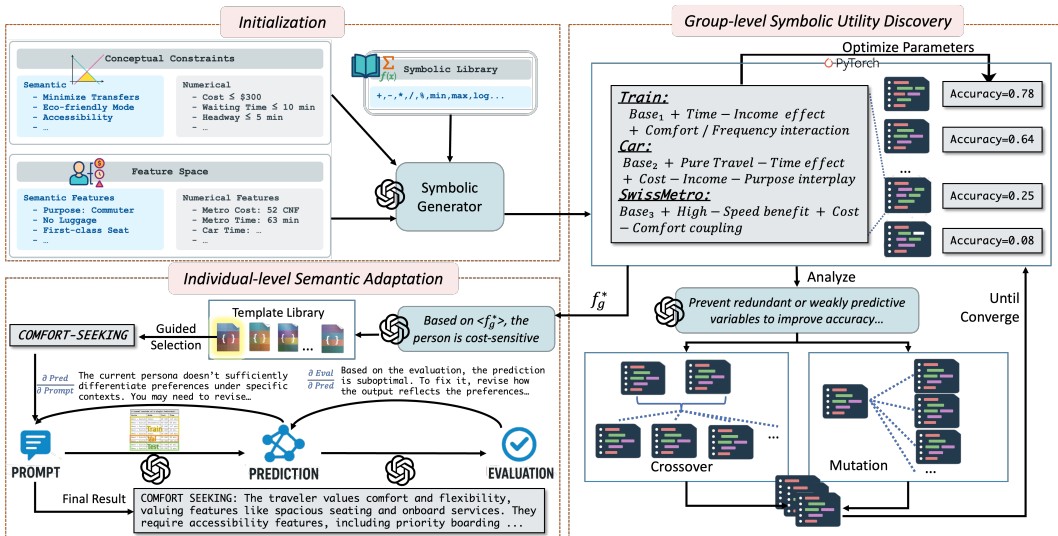

Figure 2: **ATHENA pipeline applied to a travel–mode choice example.** Here we use *Swissmetro* as an example to illustrate ATHENA framework. The *Initialization* panel encodes conceptual constraints, a mixed semantic–numerical feature space, and a symbolic library of operations. In *Group-level symbolic optimization*, an LLM samples, scores, and prunes candidate utility expressions for each alternative to produce compact formulas $\{f_g^*\}$ that best explain group behavior. In *Individual semantic adaptation*, each $f_g^*$ seeds a group-specific prompt template $\mathcal{P}_i^0$, which is refined to a personalized template via TextGrad to capture individual heterogeneity ($\mathcal{P}_i^0 \rightarrow \mathcal{P}_i^*$).

age, and income), then sampled approximately balanced subsets across these strata from the full dataset. This ensures *(i)* comparable class priors between training and test splits, and *(ii)* that no demographic group dominates the symbolic-utility discovery process. The predefined demographic grouping follows established practice in choice modeling, supports interpretability, and improves robustness by avoiding the complexity and data requirements of latent clustering methods [103–105].

**Evaluation metrics.** We report Accuracy, F1, AUC, and Cross-Entropy (CE). CE is included because ATHENA produces probabilistic predictions over choices. A lower CE means the model assigns higher probabilities to actual choices, while F1 and AUC capture classification performance; together, they provide complementary views on accuracy and calibration.

**Models and baselines.** Both stages of ATHENA, symbolic-utility discovery and individual seman-tic adaptation, run on the `gpt-4o-mini-2024-07-18` and `gemini-2.0-flash`. To evaluate its performance, we contrasted ATHENA with three baseline groups. *(i)* LLM-based methods: a plain zero-shot method [106, 107], a zero-shot chain-of-thought method [106], a five-example few-shot method [108, 109], and TextGrad tuning [96]. *(ii)* Classical discrete-choice models: Multinomial Logit (MNL) [110], Conditional Logit (CLogit) [111], and Latent-Class MNL [112]. *(iii)* Standard machine-learning classifiers: logistic regression, random forest, XGBoost [113], a shallow two-layer MLP [114], TabNet for tabular data [115], and a fine-tuned BERT classifier [116]. This spectrum ranges from end-to-end language-model reasoning through discrete choice models to conventional predictive learners, providing a balanced reference for unique modeling capabilities.

## 4.2 Overall Performance Analysis

**Performance and insights.** As shown in Table 1, on the *Swissmetro* mode choice task, ATHENA with GPT-4o-mini notably outperforms evaluated baselines across Accuracy (Acc), F1-score (F1), and AUC. Over the strongest baseline, it achieves gains of at least $6\%$ in Acc and $6.5\%$ in F1, respectively. Similar improvements are noted on the *Vaccine* dataset. Notably, our proposed method exhibits higher Cross-Entropy (CE) compared to baselines such as XGBoost. We attribute this to the inherent design of ATHENA, which produces more conservative probability distributions rather than extreme certainties. Specifically, unlike models that might predict a choice with $> 90\%$ confidence, ATHENA's framework is less prone to such high probabilities. This characteristic may better reflect the

Table 1: Performance comparison of LLM-based, classical choice, and machine learning methods on the three-class Swissmetro and three-class COVID-19 Vaccine choice tasks.

| | Method | LLM Model | Swissmetro | | | | Vaccine | | | |
|---|---|---|---|---|---|---|---|---|---|---|
| | | | Acc.↑ | F1.↑ | CE.↓ | AUC.↑ | Acc.↑ | F1.↑ | CE.↓ | AUC.↑ |
| **LLM-Based** | Zeroshot | gemini-2.0-flash | 0.5920 | 0.2940 | 0.9257 | 0.6561 | 0.5800 | 0.5092 | 0.8328 | 0.7607 |
| | | GPT-4o-mini | 0.6300 | 0.2757 | 2.7258 | 0.3657 | 0.5433 | 0.5387 | 0.8562 | 0.7395 |
| | Zeroshot-CoT | gemini-2.0-flash | 0.5880 | 0.3478 | 0.9415 | 0.6331 | 0.5800 | 0.5073 | 0.8436 | 0.7526 |
| | | GPT-4o-mini | 0.6420 | 0.2960 | 0.8957 | 0.6237 | 0.5500 | 0.5353 | 0.8540 | 0.7465 |
| | Fewshot | gemini-2.0-flash | 0.7580 | 0.7027 | 8.7244 | 0.7956 | 0.5667 | 0.5740 | 12.0324 | 0.7053 |
| | | GPT-4o-mini | 0.6815 | 0.4945 | 7.0029 | 0.7395 | 0.5067 | 0.5097 | 6.6110 | 0.6891 |
| | TextGrad | gemini-2.0-flash | 0.5568 | 0.2980 | 1.2011 | 0.5400 | 0.4241 | 0.4014 | 5.7813 | 0.6363 |
| | | GPT-4o-mini | 0.6500 | 0.3620 | 0.9079 | 0.5364 | 0.5084 | 0.4962 | 4.5412 | 0.6709 |
| | **ATHENA** | gemini-2.0-flash | 0.7679 | 0.7222 | 0.9041 | 0.8387 | 0.6797 | 0.5968 | 0.7610 | 0.8370 |
| | | GPT-4o-mini | **0.8134** | **0.7655** | 1.0863 | **0.8825** | **0.7345** | **0.7161** | 0.7551 | **0.8704** |
| **Utility Theory** | MNL | / | 0.6101 | 0.3887 | 0.8353 | 0.7074 | 0.4150 | 0.1955 | 1.0510 | 0.4301 |
| | CLogit | / | 0.5714 | 0.2424 | 0.8916 | 0.5976 | 0.4150 | 0.1955 | 1.0510 | 0.5000 |
| | Latent Class MNL | / | 0.6101 | 0.3967 | 0.8175 | 0.7182 | 0.1950 | 0.1088 | 1.0986 | 0.5000 |
| **Machine Learning** | Logistic Regression | / | 0.5620 | 0.5570 | 0.9310 | 0.7460 | 0.6500 | 0.6690 | 0.7630 | 0.8330 |
| | Random Forest | / | 0.7100 | 0.7050 | 0.7380 | 0.8810 | 0.6300 | 0.6470 | **0.7290** | 0.8420 |
| | XGBoost | / | 0.7080 | 0.7050 | 0.7040 | 0.8810 | 0.6300 | 0.6480 | 1.1420 | 0.8150 |
| | BERT | / | 0.7246 | 0.4994 | **0.7037** | 0.8811 | 0.6350 | 0.6541 | 0.7409 | 0.8168 |
| | TabNet | / | 0.6375 | 0.4060 | 0.7887 | 0.8810 | 0.6650 | 0.6684 | 0.8968 | 0.8147 |
| | MLP | / | 0.6475 | 0.6386 | 0.7626 | 0.8350 | 0.6068 | 0.6062 | 0.9320 | 0.8205 |

uncertain nature of human decision-making, which our model is designed to accommodate. Overall, the performance enhancements highlight ATHENA's strength in combining symbolic structures with semantic adaptation for effective personalized decision modeling.

**Disentangling Semantically Similar Choices.** Prompt-only LLMs and classical choice models frequently fail to distinguish between superficially similar options. For example, the few-shot LLM misclassified 75% of true *Car* trips as the premium *Metro* service. By introducing symbolic-level structure and performing individual-level semantic adaptation, ATHENA more than doubled the number of correctly classified *Car* trips, while maintaining high recall for both *Train* and *Metro*. On the Vaccine task, its learned templates encode key interactions such as age-risk trade-offs and prior-infection hesitancy, allowing it to achieve the highest F1 score despite strong semantic similarity between fully vaccinated and booster options. In practice, these interpretable templates enable a better understanding of individual behavior, for instance, identifying who tends to decline vaccination and why, which is crucial for informing high-stakes decision-making. See Appendix A.3 for details.

**Computational Complexity and Scalability.** With $T$ and $T'$ fixed, ATHENA's runtime is linear in the number of groups $|\mathcal{G}|$ and individuals $N$, scaled by the average LLM latency $\tau$:

$$\mathcal{O}((|\mathcal{G}|KT + NT')\,\tau_{\text{tok}}).$$

Both stages parallelize naturally, as group-level searches run independently and individual-level refinements can be batched or distributed. Detailed runtime measurements are provided in Appendix D.

**Extended backbone LLM comparisons.** On a 100-individual subset, we also tested larger reasoning LLMs (Qwen3-32B, DeepSeek-R1-Distill-Qwen-32B, GPT-4o). With prompt-only baselines, larger reasoning models occasionally yield higher F1/Acc but exhibit volatile calibration (high CE), reflecting the lack of structural constraints. Under ATHENA, backbone differences shrink: the symbolic discovery plus semantic adaptation turns the task into constrained sampling and small, directed improvements, allowing lightweight models to reach near-maximal performance, while stronger reasoning models provide modest, consistent gains on harder interactions (e.g., vaccine risk–trust trade-offs). Full experimental details appear in Appendix C for completeness.

## 4.3 Ablation Study

We evaluate ATHENA's two components by *(i)* keeping only the group-level symbolic utility discovery and *(ii)* keeping only the individual-level semantic adaptation, under identical data and metrics. We do not include a symbolic-only group-level discovery baseline (Stage 1 without LLM), because the Concept Library is accessible only via the LLM. Excluding it would reduce the hypothesis space to symbolic operators alone, changing the problem definition rather than providing a clean ablation.

**Group-Level Symbolic Utility Discovery: necessary but not sufficient.** When ATHENA retains only the group-level symbolic component, accuracy exceeds the classical MNL by 4.7% on *Swissmetro*

and $19\%$ on *Vaccine* (Table 2), indicating that only LLM-generated utility expressions can already encode broad demographic regularities. The accuracy trajectories of this symbolic discovery process over 30 iterations (Fig. 3) further demonstrate its effectiveness, illustrating the gradual learning of these group-level trends. Nevertheless, lower F1 score and AUC and elevated cross-entropy, reflecting limited discriminative capacity for similar alternatives. These results highlight the symbolic stage's strength in pruning the hypothesis space to interpretable structures, but also expose its limitations in capturing much heterogeneity.

Table 2: Component-wise ablation results on the Swissmetro and Vaccine choice tasks, comparing Symbolic Utility Discovery only, Semantic Adaptation only, MNL, and the full ATHENA pipeline.

| | | Swissmetro | | | | Vaccine | | | |
|---|---|---|---|---|---|---|---|---|---|
| | **Variant** | **Acc.↑** | **F1.↑** | **CE.↓** | **AUC.↑** | **Acc.↑** | **F1.↑** | **CE.↓** | **AUC.↑** |
| **Ablation Variants** | Symbolic Utility Discovery Only | 0.6566 | 0.3785 | 2.6044 | 0.5687 | 0.6067 | 0.3596 | 1.0410 | 0.7294 |
| | Semantic Adaptation Only | 0.6044 | 0.4950 | 2.2897 | 0.6872 | 0.5433 | 0.5348 | 0.8695 | 0.7535 |
| | MNL | 0.6101 | 0.3967 | 0.8175 | 0.7182 | 0.4150 | 0.1955 | 1.0510 | 0.5000 |
| | **Full Pipeline** | **0.8134** | **0.7655** | **1.0863** | **0.8825** | **0.7345** | **0.7161** | **0.7551** | **0.8704** |

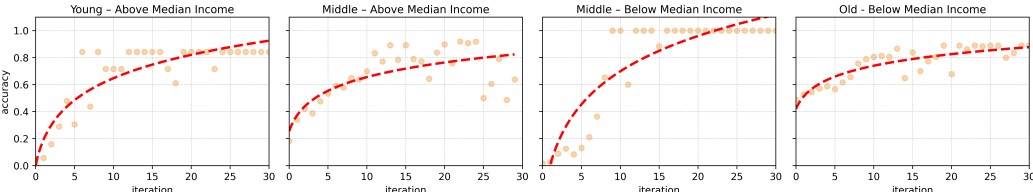

Figure 3: **Accuracy trajectories of symbolic regression.** As shown here, the accuracy keeps growing in 30 iterations for all four groups in the *vaccine* dataset. Each orange dot is the average accuracy at a given iteration; the red dashed curve is a fit showing the overall upward trend and convergence.

**Individual-Level Semantic Adaptation: powerful only with a solid starting point.** Conversely, bypassing symbolic discovery and initiating TextGrad from random templates leads to noteworthy degraded performance: As shown in Table 2, accuracy drops to $60.4\%$ on the Swissmetro dataset and $54.3\%$ on the Vaccine dataset; Swissmetro's CE more than doubles (2.29), and AUC falls below $0.70$. Without a sound starting point, gradients are likely to converge to local optima and yield erratic probability outputs, reaffirming the unreliability of unguided adaptation in multi-choice settings.

**Take-away.** The two stages of ATHENA are complementary: symbolic discovery supplies an interpretable, well-regularized search space, while semantic adaptation injects the individual-level nuance that symbolic rules alone miss.

### 4.4 Symbolic Utility Discovery Fragment Analysis

Equation (8) shows that an individual prediction is influenced by group-level symbolic utility $f_g^*(X_i; \theta_g^*)$. In this section, we demonstrate the building blocks of those utilities are both behaviorally meaningful and reusable across groups. As shown in Fig. 4, each symbolic utility is decomposed into atomic fragments $\{\varphi_1, \varphi_2, \dots\}$ and their global importance is quantified.

**Fragment score.** For every group $g$ we retain the top-K ($K = 3$) utilities ranked by held-out accuracy $\text{Acc}(f)$. The importance score of a fragment $\varphi_m$ is then

$$\text{Score}(\varphi_m) = \sum_{g \in \mathcal{G}} \sum_{k=1}^{K} \mathbb{1}\left[\varphi_m \subset \left\{f_{g,k}^*\right\}\right] \cdot \text{Acc}\left(f_{g,k}^*\right) \tag{9}$$

So a fragment earns points whenever it (i) appears in the top-ranked utilities of *many* groups and (ii) is embedded in highly predictive expressions.

Fig. 4 visualizes the fragment scores for both datasets. Only a small fraction of fragments dominate, confirming that ATHENA converges to a compact and interpretable symbolic basis. For example, in *Vaccine*, one of the leading fragment $\varphi_7 = \sqrt{\text{Age}} * (\text{Trust Government} + \text{Trust\_Science})$ softens

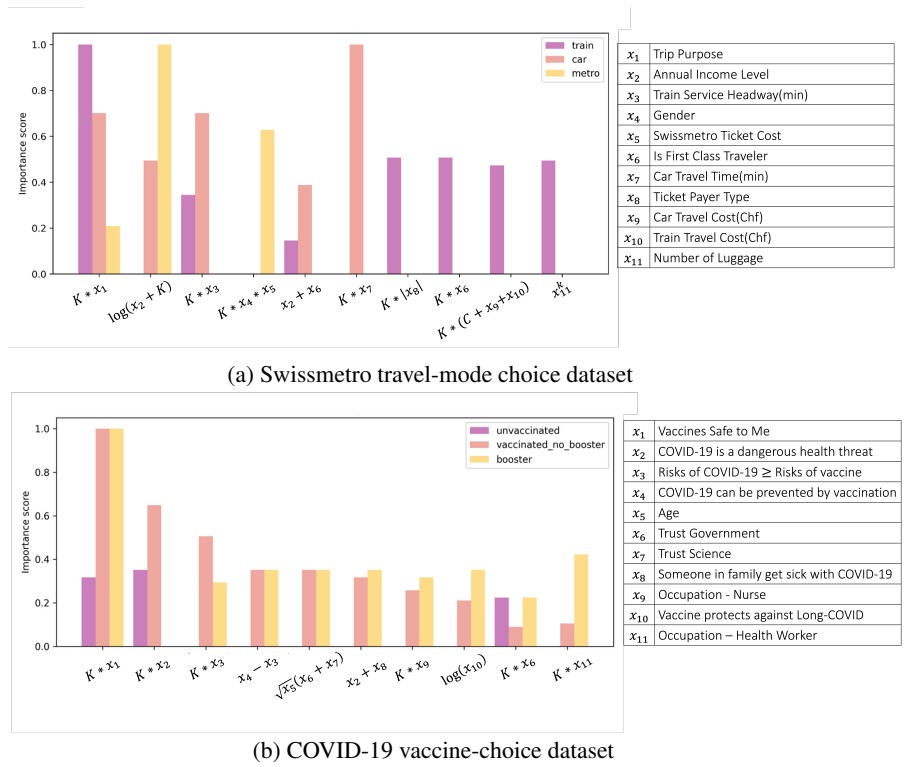

(a) Swissmetro travel-mode choice dataset

(b) COVID-19 vaccine-choice dataset

Figure 4: **Aggregated fragment importance extracted from the learned symbolic utilities.** For each task, we plot the top-ranked atomic fragments $\varphi_m$ that appear in the three best group-level utility formulas and weight them by fragment importance score 9. Values shown here are the normalized scores in $[0, 1]$.

age's impact at higher values while amplifying it for individuals who trust government or science, precisely isolating the cohort most likely to take boosters.

Beyond fragment-level analysis, ATHENA also produces fully interpretable symbolic utilities. Representative full formulas and domain-relevant insights for both *Swissmetro* and *Vaccine* are provided in Appendix A.4.

## 5   Conclusion

This research highlights the critical role of textual-semantic information in overcoming the limitations of traditional utility-based models for human decision-making. By introducing ATHENA, an adaptive textual-symbolic and human-centric reasoning framework is proposed that integrates group-level symbolic regression of utility functions with individual-level, LLM-powered semantic modeling, we offer a more comprehensive and personalized view of choice behavior. Our experiments on transportation mode choice and vaccine uptake demonstrate that this co-design approach clearly outperforms three existing model zoos, including classical utility, machine learning, and purely LLM-based approach, underscoring the benefits of capturing both structured attributes and rich semantic context. These findings suggest that textualized-symbolic reasoning can bridge the gap between theoretical utility optimization and real-world individual choices, paving the way for more adaptive and human-centric decision models.

**Limitations.** The current implementation of ATHENA has two limitations. 1) Computational Complexity: The proposed framework requires extra computational resources for textual gradient, particularly when scaling to larger populations. 2) Representation on Groups: ATHENA assumes that a shared symbolic utility function can effectively model each demographic group. However, groups with greater internal diversity may produce weaker or less reliable representations. 3) Result Stability: All reported results are based on single representative runs under fixed random seeds, given the computational cost of multi-stage adaptation. Future work will include multi-seed repetitions to further examine the stability of ATHENA's performance.

## Acknowledgments

Yang Zhao acknowledges a fellowship from JHU + Amazon Initiative for Interactive AI.

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

# A Extended Analysis

## A.1 Broader Societal Impacts

The introduction of ATHENA, a two-stage framework that first discovers group-level symbolic utility functions and then develops individual semantic templates, offers potential for positive societal contributions. By combining LLM reasoning with symbolic regression, ATHENA aims to deliver: **(1) More inclusive public-policy insights:** interpretable utility functions reveal group motivations behind choices in socially critical behaviors, enabling better targeted interventions and policy making. **(2) Precise and equitable individual adaptation:** reduce the "one-size-fits-all" errors in some high-stake domains, like healthcare and education.

At the same time, ATHENA also brings concerns. (1) Potential Misuse: Because ATHENA can model fine-grained individual decisions, it could be deployed for manipulative advertising, political micro-targeting, or discriminatory dynamic pricing. Mitigations include restricted licensing and mandatory human oversight for high-impact deployments. (2) Bias and Fairness: The symbolic utility discovery stage may suffer from coarse group partitioning, which can ignore intra-group heterogeneity. Meanwhile, the semantic adaptation stage may inherit biases presented in the LLM's training data, thereby embedding historical stereotypes.

## A.2 Qualitative Analysis and Case Study

Table 3: **Qualitative case study of ATHENA on the *Swissmetro* dataset.** This table contrasts four representative travelers' attributes, the candidate alternatives, the group-level symbolic utility functions learned in Stage-1, and the individualized decision rules refined in Stage-2.

| *Swissmetro* Case 1 | |
| --- | --- |
| Features $\mathcal{X}$ | - **Age**: between 54 and 65 years old
- **Gender**: Male
- **Income**: Over 100
- **Trip Purpose**: business
- **Luggage**: none of luggage
- **Payment Method**: paid by oneself
- **Origin**: St. Gallen
- **Destination**: Bern |
| Alternatives $\mathcal{J}$ | - **Metro**: travel time of 77 minutes, costing 74 CHF, with a headway of 30 minutes
- **Train**: travel time of 120 minutes, costing 64 CHF, with a headway of 120 minutes
- **Car**: travel time of 169 minutes, costing 60 CHF |
| Optimal Group Utility $f_g^*$ | - **Train**: $K_1 \cdot \text{trip\_purpose} + K_2 \cdot (\sqrt{\text{num\_luggage} + C_1} + C_2) \cdot \log(\text{age} + C_3) + K_3 \cdot \text{time\_train} + K_4 \cdot \sqrt{\text{train\_headway\_min}} + K_5 \cdot (\text{is\_first\_class} + C_4) \cdot \sqrt{\log(\text{income} + C_5)} + C_6$
- **Car**: $K_1 \cdot \text{trip\_purpose} + K_2 \cdot \sqrt{\text{num\_luggage} + C_1} \cdot \log(\text{age} + C_2) + K_3 \cdot \text{time\_car} \cdot \sqrt{\log(\text{income} + C_3)} + K_4 \cdot \sqrt{|\text{cost\_train} + C_4|} + K_5 \cdot \text{train\_service\_headway} + K_6 \cdot (\text{is\_first\_class} + C_5) \cdot \sqrt{\log(\text{income} + C_6)} + C_7$
- **Metro**: $K_1 \cdot \text{trip\_purpose} + K_2 \cdot (\text{num\_luggage} + C_1) \cdot \sqrt{\log(\text{age} + C_2)} + K_3 \cdot \text{time\_sm} \cdot \sqrt{\text{income} + C_3} + K_4 \cdot \sqrt{\text{cost\_sm} + C_4} + K_5 \cdot (\text{ticket\_payer\_type} + C_5) + C_6$ |
| Optimal Personalized Decision Rule $\mathcal{P}_i^*$ | - **BALANCED**: As a business traveler, my travel decisions are guided by a dynamic scoring system that prioritizes speed, environmental impact, and cost. I often weigh these factors differently based on specific scenarios (e.g., major conferences vs. regular trips). Real-time traffic/weather and reflections on past trips refine preferences; I consider Swissmetro, trains, cars, buses, and rideshares, evaluating comfort, reliability, and amenities. |

| | |
|---|---|
| *Swissmetro – Case 2* | |
| Features $\mathcal{X}$ | - **Age**: between 39 and 54 years old
- **Gender**: Male
- **Income**: between 50 and 100
- **Trip Purpose**: leisure
- **Luggage**: no luggage
- **Payment Method**: paid by oneself
- **Origin**: Graubünden
- **Destination**: Bern |
| Alternatives $\mathcal{J}$ | - **Metro**: travel time 142 min, cost 123 CHF, headway 30 min
- **Train**: travel time 180 min, cost 97 CHF, headway 60 min
- **Car**: travel time 136 min, cost 149 CHF |
| Optimal Group Utility $f_g^*$ | - **Train**: $K_1 \cdot (\text{ticket\_payer\_type} \cdot \text{is\_car\_available}\sqrt{\text{age} + C_1} + K_2 \cdot \text{is\_first\_class} \cdot \log(\text{age} + C_2)) - K_3 \cdot \dfrac{\text{time\_train}}{\text{age} + C_3} + K_4 \cdot (\text{income} + \text{num\_luggage}\sqrt{\text{age} + C_4}) - K_5 \cdot \left(\dfrac{\text{time\_car}}{\text{age} + C_5} + C_6\right) + C_7$

- **Car**: $K_1 \cdot (\text{ticket\_payer\_type} \cdot \text{is\_car\_available}\sqrt{\text{age} + C_1} + K_2 \cdot \text{is\_first\_class}) - K_3 \cdot \dfrac{\text{time\_train}}{\text{age} + C_2} + K_4 \cdot \log(\text{income} + \text{num\_luggage} + K_5\sqrt{\text{age} + C_3}) - K_6 \cdot \left(\dfrac{\text{time\_car}}{\text{age} + C_4} + C_5\right) + C_6$

- **Metro**: $K_1 \cdot (\text{sm\_headway\_min} - \dfrac{\text{time\_sm}}{\log(\text{age} + C_1)}) + K_2 \cdot (\text{has\_ga\_travel\_pass}\sqrt{\text{age} + C_2} + \text{gender} \cdot \log(\text{income} + \text{num\_luggage} + C_3)) + C_4$ |
| Optimal Personalized Decision Rule $\mathcal{P}_i^*$ | - **BALANCED, INFORMED, SUSTAINABLE, USER-CENTRIC, CONTEXT-ADAPTIVE, COMFORT-FOCUSED**: Dynamically trades off time, cost, and scenic enjoyment. Eco-friendliness and comfort (leg-room, quiet cars) matter, even with longer trips or higher prices; prior-trip feedback and real-time context refine recommendations. |
| *Swissmetro – Case 3* | |
| Features $\mathcal{X}$ | - **Age**: between 39 and 54 years old
- **Gender**: Male
- **Income**: under 50
- **Trip Purpose**: leisure
- **Luggage**: no luggage
- **Payment Method**: paid by unknown people
- **Origin**: Zurich
- **Destination**: Bern |
| Alternatives $\mathcal{J}$ | - **Metro**: travel time 56 min, cost 42 CHF, headway 10 min
- **Train**: travel time 111 min, cost 36 CHF, headway 30 min
- **Car**: travel time 88 min, cost 60 CHF |

| | |
|---|---|
| Optimal Group Utility $f_g^*$ | - **Train**: $C_1 + K_1 \cdot \|\text{time\_train} - \text{time\_car}\|(\log(\text{income} + C_2) + \sqrt{\text{age} + C_3})(\text{gender}\sqrt{\text{num\_luggage} + C_4} + \text{is\_first\_class}) + K_2 \cdot \dfrac{\text{trip\_purpose}}{\sqrt{\text{age} + C_5} + C_6} - K_3 \cdot \text{cost\_train} \cdot \text{trip\_purpose}$

- **Car**: $C_1 + K_1 \cdot (\text{num\_luggage} \cdot \text{trip\_purpose} \cdot \|\log(\text{income} + C_2)\| \cdot (C_3 + \sqrt{\text{age} + C_4}) + \sqrt{\text{age} + C_5} \cdot \|\text{time\_train} - \text{time\_car}\| \cdot (C_6 + \text{is\_first\_class})) - K_2 \cdot (C_7 + \text{train\_service\_headway\_min}) + K_3 \cdot (\text{income} + C_8)^{C_9}$

- **Metro**: $C_1 + K_1 \cdot \left( \dfrac{\text{time\_sm}^{C_2} \cdot \text{trip\_purpose} \cdot \sqrt{\text{num\_luggage} + C_3} \cdot \log(\text{income} + C_4)}{\sqrt{\text{age} + C_5} + C_6} + K_2 \cdot (\text{cost\_sm} \cdot (\log(\text{age} + C_7) + \text{is\_first\_class})) \cdot (C_8 + \sqrt{\text{time\_car}}) \right)$ |
| Optimal Personalized Decision Rule $\mathcal{P}_i^*$ | - **COST_SAVING**: Prefers options at least 10% cheaper than peak-fare averages, values minimal transfers and amenities (Wi-Fi, food), uses carbon-footprint data, and favors off-peak scheduling; feedback refines future recommendations. |

| *Swissmetro – Case 4* | |
|---|---|
| Features $\mathcal{X}$ | - **Age**: over 65 years old
- **Gender**: Male
- **Income**: over 100
- **Trip Purpose**: shopping
- **Luggage**: no luggage
- **Payment Method**: paid half-half
- **Origin**: Vaud
- **Destination**: Geneva |
| Alternatives $\mathcal{J}$ | - **Metro**: travel time 21 min, cost 226 CHF, headway 10 min
- **Train**: travel time 42 min, cost 209 CHF, headway 60 min
- **Car**: travel time 40 min, cost 67 CHF |
| Optimal Group Utility $f_g^*$ | - **Train**: $K_1 \cdot (\text{time\_train} + \text{num\_luggage} \cdot \text{age}^{C_1} + \text{age}^{C_2} + \text{time\_car}) + K_2 \cdot (\text{income} \cdot \text{is\_first\_class} \cdot \text{gender} \cdot \log(\text{age} + C_3)) + \text{num\_luggage}^{C_4} + C_5$
- **Car**: $K_1 \cdot (\text{time\_car} + \text{num\_luggage} \cdot \text{age}^{C_1} + \text{income}\sqrt{\text{is\_first\_class}} + \text{car\_travel\_cost\_chf} + \text{num\_luggage} \cdot \exp(\text{age}/C_2)) + C_3$
- **Metro**: $K_1 \cdot \text{time\_sm} + K_2 \cdot (\text{cost\_sm} + \text{income} \cdot (\text{is\_first\_class} + \text{gender}) \cdot \exp(\text{age}^{C_1}) + \text{num\_luggage}\,\text{age}^{C_2} + \text{age}/C_3) + C_4$ |
| Optimal Personalized Decision Rule $\mathcal{P}_i^*$ | - **COMFORT_SEEKING**: Prioritizes spacious seating, quiet cars, and onboard services; willing to pay up to 20% premium. Prefers real-time updates and easy boarding for accessibility; social events may nudge to more social modes; feedback refines future recommendations. |

Table 4: **Qualitative case study of ATHENA on the *Vaccine* dataset.** This table contrasts four representative individuals' attributes, the candidate alternatives, the group-level symbolic utility functions learned in Stage-1, and the individualized decision rules refined in Stage-2.

| *Vaccine – Case 1* | |
|---|---|

| | |
|---|---|
| Features $\mathcal{X}$ | - **Age**: 25
- **Gender**: Male
- **Occupation**: Nurse
- **Education**: No university degree
- **Income**: Above-median
- **COVID-19 Threat Perception**: Moderate
- **Risk Perception**: Disease risk > vaccine risk
- **Trust in Government**: Moderate
- **Trust in Science**: Moderate
- **Perceived Vaccine Safety**: Fairly safe
- **Family COVID Infection**: >1 yr ago
- **Attention to Vaccine News**: Increased |
| Alternatives $\mathcal{J}$ | - **Unvaccinated**
- **Vaccinated_No_Booster**
- **Booster** |
| Optimal Group Utility $f_g^*$ | - **Unvaccinated**: $C_1 \cdot \text{covid\_threat}(C_2 + \text{trust\_gov} \cdot \text{trust\_sci} \cdot \log(\text{age} + C_3)) \cdot \text{risk\_covid\_gt\_vax} + K_1 \cdot \text{family\_covid} \cdot \log(\text{age} + C_4)$
- **Vaccinated_No_Booster**: $C_1 \cdot \text{covid\_threat} + C_2 \cdot \text{vax\_safe} + K_1 \cdot (\text{trust\_gov} \cdot \text{trust\_sci} \cdot \text{more\_attention} \sqrt{\text{age} + C_3})$
- **Booster**: $C_1 \cdot e^{\text{age}^{C_2}} \text{covid\_threat} \sqrt{\text{vax\_protect\_long}} + C_3 \cdot \text{vax\_safe} + K_1 \cdot (\text{trust\_gov} \cdot \text{trust\_sci} \cdot \text{nurse} \cdot \sqrt{\text{age} + C_4})$ |
| Optimal Personalized Decision Rule $\mathcal{P}_i^*$ | - **TRUSTING_AUTHORITY**: This persona represents a cautiously informed healthcare worker who values evidence-based guidance and may favor "Vaccinated (No Booster)" given safety concerns, while remaining open to updates as new data emerge; family and social influence are considered. |
| *Vaccine – Case 2* | |
| Features $\mathcal{X}$ | - **Age**: 55
- **Gender**: Male
- **Education**: No university degree
- **Income**: Below-median
- **COVID-19 Threat Perception**: Moderate
- **Trust in Government Delivery**: High
- **Trust in Science**: Some
- **Risk Perception**: Disease risk > vaccine risk
- **Family COVID**: None
- **Attention to Vaccine News**: Decreased |
| Alternatives $\mathcal{J}$ | - **Unvaccinated**
- **Vaccinated_No_Booster**
- **Booster** |
| Optimal Group Utility $f_g^*$ | - **Unvaccinated**: $K_1\sqrt{\text{covid\_threat}(\text{risk\_covid\_gt\_vax} + \text{gender}\sqrt{\text{age} + C_1})} \cdot ((\text{trust\_gov}\,\text{trust\_sci})^{C_2} + C_3) + K_2 \cdot \text{more\_attention} - K_3 \cdot \text{low\_income}(C_4 + \text{has\_degree} \cdot \text{trust\_gov} \cdot \text{trust\_sci}) + C_5$
- **Vaccinated_No_Booster**:
$K_1 \cdot (\text{vax\_safe} + \text{trust\_gov} \cdot \text{trust\_sci} \cdot \sqrt{\sqrt{\text{age} + C_1} + \text{income\_unknown} + C_2} + C_3) + K_2 \cdot \dfrac{\text{more\_attention}}{\text{less\_attention} + C_4} + C_5$
- **Booster**: $K_1 \cdot (\text{family\_covid} + \text{physician} \cdot \text{trust\_gov}\,\text{trust\_sci}\,(\sqrt{\text{age} + C_1} + C_2) + \text{nurse} \cdot \text{trust\_sci}\sqrt{\text{age} + C_3} + C_4) + C_5$ |

| | |
|---|---|
| Optimal Personalized Decision Rule $\mathcal{P}_i^*$ | - **SKEPTICAL**: Prefers conservative choices due to perceived safety concerns; may remain unvaccinated unless convinced by trusted figures; open to "Vaccinated_No_Booster" or "Booster" if necessity and safety are clearly established. |

### *Vaccine – Case 3*

| | |
|---|---|
| Features $\mathcal{X}$ | - **Age**: 86
- **Gender**: Male
- **Education**: No university degree
- **Income**: Below-median
- **COVID-19 Threat View**: Moderate
- **Perceived Vaccine Safety**: High
- **Long-COVID Protection Belief**: Uncertain
- **Family COVID**: None
- **Attention to Vaccine News**: Increased |
| Alternatives $\mathcal{J}$ | - **Unvaccinated**
- **Vaccinated_No_Booster**
- **Booster** |
| Optimal Group Utility $f_g^*$ | - **Unvaccinated**:
$C_1 + K_1 \cdot (\text{covid\_threat} \cdot \text{trust\_gov} \cdot \text{trust\_sci} \cdot \text{age}^{C_2}(C_3 + \text{family\_covid})) - K_2 \cdot (\text{risk\_covid\_gt\_vax}(C_4 + \text{income\_unknown} \cdot e^{-K_3 \cdot \text{more\_attention age}^{C_5}}))$
- **Vaccinated_No_Booster**: $C_1 - K_1 \cdot (\text{age}^{C_2}(C_3 + \text{low\_income})(C_4 - \text{trust\_sci})) + K_2 \cdot (\text{vax\_safe trust\_gov}(C_5 + \text{more\_attention age}^{C_6}))$
- **Booster**: $C_1 + K_1 \cdot (\text{vax\_protect\_long} \cdot e^{-K_2(\text{age}^{C_2} + \text{low\_income} \cdot \text{family\_covid})}) - K_3 \cdot (\text{less\_attention} \cdot \text{trust\_sci}(\text{age}/C_3))$ |
| Optimal Personalized Decision Rule $\mathcal{P}_i^*$ | - **THREAT_AVOIDING**: Perceives high disease risk; ranks *Booster > Vaccinated_No_Booster > Unvaccinated*; considers logistics and side-effect concerns while relying on trusted sources. |

### *Vaccine – Case 4*

| | |
|---|---|
| Features $\mathcal{X}$ | - **Age**: 52
- **Gender**: Male
- **Education**: No university degree
- **Income**: Below-median
- **COVID-19 Threat Perception**: Strong
- **Belief in Vaccine Prevention**: Low
- **Trust in Science**: Moderate
- **Risk Perception**: Disease risk > vaccine risk
- **Attention to Vaccine News**: Unchanged
- **Family COVID**: None |
| Alternatives $\mathcal{J}$ | - **Unvaccinated**
- **Vaccinated_No_Booster**
- **Booster** |

| | |
|---|---|
| Optimal Group Utility $f_g^*$ | - **Unvaccinated**: $K_1\sqrt{\text{covid\_threat}}(\text{risk\_covid\_gt\_vax} + \text{gender}\sqrt{\text{age}} + C_1) \cdot ((\text{trust\_gov trust\_sci})^{C_2} + C_3) + K_2 \cdot \text{more\_attention} - K_3 \cdot \text{low\_income}(C_4 + \text{has\_degree} \cdot \text{trust\_gov} \cdot \text{trust\_sci}) + C_5$
- **Vaccinated_No_Booster**: $K_1 \cdot (\text{vax\_safe} + \text{trust\_gov} \cdot \text{trust\_sci} \cdot \sqrt{\sqrt{\text{age} + C_1} + \text{income\_unknown} + C_2} + C_3) + K_2 \cdot \dfrac{\text{more\_attention}}{\text{less\_attention} + C_4} + C_5$
- **Booster**: $K_1 \cdot (\text{family\_covid} + \text{physician} \cdot \text{trust\_gov} \cdot \text{trust\_sci}(\sqrt{\text{age} + C_1} + C_2) + \text{nurse} \cdot \text{trust\_sci}\sqrt{\text{age} + C_3} + C_4) + C_5$ |
| Optimal Personalized Decision Rule $\mathcal{P}_i^*$ | - **BALANCED**: Cautious yet data-driven; moderate trust in authorities; open to boosters with clear evidence; weighs prior experiences and accessibility. |

## A.3 Semantically Similar Choices Analysis

As illustrated in Figure 5, ATHENA not only raises aggregated accuracy but also improves decision-critical boundaries, offering more reliable evidence for public-health and transport-policy planning.

## A.4 Extended Interpretability Showcase

We provide representative full symbolic utilities discovered by ATHENA on *Swissmetro* and *Vaccine* datasets. These examples illustrate how the symbolic structure translates into actionable insights for transportation and public health domains.

### A.4.1 Representative Example — Swissmetro Dataset

| Mode | Discovered symbolic utility |
|---|---|
| **Train** | $K_1 \cdot (\text{train\_time} + \text{metro\_time} + \text{luggage} \cdot \log(\text{age} + C_1) + \text{age} + \text{is\_male}) + C_2 \cdot (\text{first\_class} + \text{income}) - C_3 \cdot (\text{GA\_pass} + \text{headway})$ |
| **Car** | $K_1 \cdot (\text{car\_time} + \text{train\_time} + \text{luggage} \cdot \log(\text{age} + C_1) + \text{age}) + C_2 \cdot (\text{first\_class} + \text{income}) - C_3 \cdot (\text{GA\_pass} + \text{metro\_fare} + \text{is\_male})$ |
| **Metro** | $K_1 \cdot (\text{metro\_time} + \text{luggage} + \text{age} + \text{is\_male}) + C_2 \cdot (\text{first\_class} + \text{income}) - C_3 \cdot (\text{headway} + \text{GA\_pass} + \text{is\_male})$ |

**Feature:** Between 39 and 54 years old, identify as female, and have an income between 50 and 100.

**Key take-aways for domain experts**

- **Time dominates.** Large negative coefficients on travel-time variables show this segment is **highly time-sensitive** → investments that shorten door-to-door time (e.g., skip-stop service) should shift demand [117].

- **Comfort premium.** Positive weight on (first_class + income) across all modes indicates a willingness to pay for comfort that scales with income → targeted upselling (seat reservations, quiet cars) is effective [118].

- **Luggage burden grows with age.** The interaction luggage $\cdot \log(\text{age} + C_1)$ reveals baggage becomes disproportionately painful for older travelers → facilities such as luggage trolleys or porter services may raise train/metro share [119].

- **GA pass effect.** Owning a GA pass biases travellers away from modes that still incur extra fares. Extending GA coverage to Swissmetro would therefore raise its relative appeal [120].

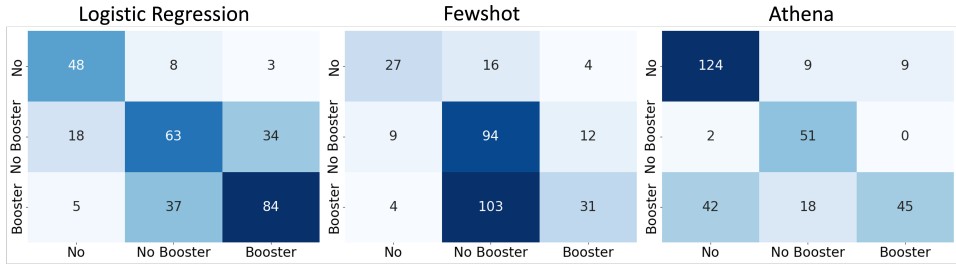

(a) **Vaccine-uptake task.** ATHENA removes all *34* cases in which the *Vaccinated_no_booster* class was previously misclassified as *Booster*, thereby preserving the integrity of booster-demand estimates.

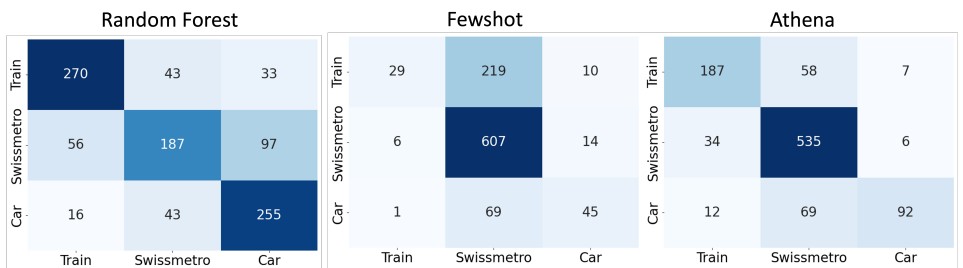

(b) **Travel-mode choice task.** ATHENA cuts the *Swissmetro*-versus-*Car* confusion from *83* to *6* instances, refining forecasts of low-carbon rail adoption.

Figure 5: ATHENA yields improvements on the classes that matter most yet were previously hard to distinguish.

| Mode | Discovered symbolic utility |
|---|---|
| **Train** | $K_1 \cdot \Big( \text{purpose} + |\text{payer\_type} \cdot C_1| - \text{first\_class} + |\text{luggage}| \sqrt{|\text{age} + C_2|} + |\text{train\_time} + C_3| + \log(\text{income} + C_4) \Big) - C_5$ |
| **Car** | $K_1 \cdot \Big( |\text{car\_time} + C_1| + |\text{car\_time} - \text{train\_time} + C_2| - |\text{car\_cost} + \text{train\_cost} + C_3| + |\text{headway}| \sqrt{\text{income} + C_4} \Big) + C_5$ |
| **Metro** | $K_1 \cdot \Big( |\text{metro\_time} + C_1| + |\text{metro\_cost} + C_2| + \sqrt{|\text{age} + C_3|} + \log\big( \exp(\text{income} + C_4) + C_5 \big) \Big) - C_6$ |

**Feature:** Male travelers younger than 24 years, annual income 50–100k.

**Key take-aways for domain experts**

- **Time still trumps money.** Travel time appears in all utilities, while fare only in Car/Metro. For under-25 travelers, each minute lost matters more than an extra franc → prioritizing faster transfers or signal priority is especially effective [121].

- **Headway frustration fuels car use.** The term headway $\cdot \sqrt{\text{income}}$ shows that infrequent trains push young people toward cars, and irritation rises with income → higher-frequency rail services can curb car switching [122].

- **First-class indifference.** The negative first_class coefficient suggests little interest in upgrades → amenities in standard class (Wi-Fi, gaming lounges) may be more persuasive than premium seating [123].

### A.4.2 Representative Example — Vaccine Dataset

| Mode | Discovered symbolic utility |
|---|---|
| Unvaccinated | $C_1 \cdot \text{covid\_threat} \cdot \Big(C_2 + \text{trust\_government} \cdot \text{trust\_science} \cdot \log(\text{age} + C_3)\Big) \cdot \text{risk\_of\_covid\_greater\_than\_vax} + K_1 \cdot \text{have\_covid\_sick\_family\_member} \cdot \log(\text{age} + C_4)$ |
| Vaccinated (no booster) | $C_1 \cdot \text{covid\_threat} + C_2 \cdot \text{vaccine\_safe\_to\_me} + K_1 \cdot (\text{trust\_government} \cdot \text{trust\_science} \cdot \text{more\_attention\_to\_vax\_info} \cdot \sqrt{\text{age} + C_3})$ |
| Booster | $C_1 \cdot e^{\text{age}^{C_2}} \cdot \text{covid\_threat} \cdot \sqrt{\text{vax\_protect\_long\_yes}} + C_3 \cdot \text{vaccine\_safe\_to\_me} + K_1 \cdot (\text{trust\_government} \cdot \text{trust\_science} \cdot \text{nurse} \cdot \sqrt{\text{age} + C_4})$ |

**Feature:** Age 18–38, income above county median.

**Key take-aways for domain experts**

- **Risk trade-off in vaccination choice.** The product covid_threat × risk_of_covid_greater_than_vax captures a critical decision-making trade-off. Messaging must narrow this perceived risk gap, e.g., by emphasizing robust evidence on vaccine safety [124].

- **Booster demand rises steeply with age.** The factor $e^{\text{age}^{C_2}}$ generates a nonlinear age effect: as age increases, perceived vaccine benefit grows rapidly. This reflects age-associated increases in risk perception and vulnerabilities [125].

- **Prior belief and healthcare occupation.** The presence of vax_protect_long_yes and nurse occupation in the booster equation means emphasizing extended protection and occupation will push this group further along the vaccination ladder [126].

- **Trust is pivotal for vaccine uptake.** The multiplicative trust_government × trust_science term appears in every vaccinated utility, signalling that confidence in both institutions amplifies willingness [127].

| Mode | Discovered symbolic utility |
|---|---|
| Unvaccinated | $K_1 \cdot \sqrt{\text{covid\_threat}} \cdot (\text{risk\_of\_covid\_greater\_than\_vax} + \sqrt{\text{age}} \cdot \text{gender} + C_1) \cdot ((\text{trust\_government} \cdot \text{trust\_science})^2 + C_2) + K_2 \cdot \text{more\_attention\_to\_vax\_info} - K_3 \cdot (\text{income\_below\_median} \cdot \text{have\_university\_degree} \cdot (\text{trust\_government} \cdot \text{trust\_science})) + C_3$ |
| Vaccinated (no booster) | $K_1 \cdot (\text{vaccine\_safe\_to\_me} + \text{trust\_government} \cdot \text{trust\_science} \cdot \sqrt{\sqrt{\text{age} + C_1} + \text{income\_unknown} + C_2}) + K_2 \cdot \dfrac{\text{more\_attention\_to\_vax\_info}}{\text{less\_attention\_to\_vax\_info} + C_3} + C_4$ |
| Booster | $K_1 (\text{have\_covid\_sick\_family\_member} + \text{physician} (\text{trust\_government} \cdot \text{trust\_science}) \cdot (\sqrt{\text{age} + C_1} + C_2) + \text{nurse} (\text{trust\_science} \cdot \sqrt{\text{age} + C_3})) + C_4$ |

**Feature:** Adults with varied trust, income, and education profiles.

**Key take-aways for domain experts**

- **Information attention as lever.** Positive weights on more_attention_to_vax_info indicate that engagement with vaccine information consistently increases uptake → interactive campaigns remain essential [128].

- **Nonlinear trust amplification.** The squared term $(\text{trust\_government} \cdot \text{trust\_science})^2$ highlights a super-additive effect → boosting both trust dimensions together disproportionately reduces hesitancy [129].

- **Education buffers income hesitancy.** The negative income effect is mitigated by education–trust interactions → higher education plus trust can offset low-income hesitancy, pointing to education-focused outreach [130].

# B  Baseline Setup

## B.1  Utility-Based Models

Table 9: Utility-based models and key settings (train : test = 0.8 : 0.2)

| Model | Key (Non-default) Settings |
|---|---|
| SimpleMNL | `intercept="item"; optimizer="adam"` |
| ConditionalLogit | `optimizer="adam"`; added intercept for items 1 & 2 |
| Latent Class MNL | `n_latent_classes=2; fit_method="mle";` `optimizer="adam"; epochs=1000` |

## B.2  Machine Learning Models

Table 11: Machine learning models and key settings (train : test = 0.8 : 0.2)

| Model | Best hyper-parameters |
|---|---|
| Logistic Regression | `C=10, penalty=l2, solver=saga` |
| Random Forest | `bootstrap=False, max_depth=None, min_samples_leaf=1,` `min_samples_split=2, n_estimators=600` |
| XGBoost | `colsample_bytree=0.8, learning_rate=0.05, max_depth=6,` `n_estimators=500, subsample=0.8` |

## B.3  LLM-Based Models

Take **Swissmetro** dataset as an example.

---

**Prompt B.1:** *Swissmetro* **- Zeroshot**

[SYS] You are a decision assistant that predicts a probability distribution over three travel modes, Swissmetro, Train, and Car, for a single trip.
You will receive two blocks of text:
<TRIP_INFO> . . .   details like trip purpose, luggage, payment, origin, destination . . . </TRIP_INFO>
<TRANSPORT_OPTIONS> . . . list of modes with travel time, cost, headway . . . </TRANSPORT_OPTIONS>
**Instructions:**
1. Use only the information in <TRIP_INFO> and <TRANSPORT_OPTIONS>.
2. Estimate a probability for each mode so they sum to 1.
3. **Output only** a JSON object, for example:
"'json { "Swissmetro": <float between 0 and 1>, "Train": <float between 0 and 1>, "Car": <float between 0 and 1> } "'
No additional text; just the JSON object with normalized probabilities.
[USR] <TRIP_INFO> {trip_info} </TRIP_INFO>
<TRANSPORT_OPTIONS> {transport_options} </TRANSPORT_OPTIONS>

---

**Prompt B.2:** *Swissmetro* - **Zeroshot-CoT**

[SYS] You are a decision assistant that predicts a probability distribution over three travel modes, Swissmetro, Train, and Car, for a single trip.
You will receive two blocks of text:
<TRIP_INFO> . . .   details like trip purpose, luggage, payment, origin, destination . . . </TRIP_INFO>
<TRANSPORT_OPTIONS> . . . list of modes with travel time, cost, headway . . . </TRANSPORT_OPTIONS>
**Instructions:**
1. Use only the information in <TRIP_INFO> and <TRANSPORT_OPTIONS>.
2. Estimate a probability for each mode so they sum to 1.
3. **Output only** a JSON object, for example:
"'json { "Swissmetro": <float between 0 and 1>, "Train": <float between 0 and 1>, "Car": <float between 0 and 1> } "'
No additional text; just the JSON object with normalized probabilities.
Let's think step-by-step.
[USR] <TRIP_INFO> {trip_info} </TRIP_INFO>
<TRANSPORT_OPTIONS> {transport_options} </TRANSPORT_OPTIONS>

**Prompt B.3:** *Swissmetro* **- Fewshot**

[SYS] You are a decision assistant that predicts a probability distribution over three travel modes—Swissmetro, Train, and Car—for a set of travel records.

You will receive multiple records. Each record consists of three blocks: <TRIP_INFO> … trip details: purpose, luggage, payment, origin, destination … </TRIP_INFO> <TRANSPORT_OPTIONS> … each mode's travel time, cost, headway … </TRANSPORT_OPTIONS> <CHOICE> … either a JSON object with probabilities (for examples), or left empty for the record to predict … </CHOICE>

**Instructions:** - For records where <CHOICE> is filled, treat them as examples. - For the final record (with an empty <CHOICE>), output **only** the JSON object of normalized probabilities (summing to 1), with no extra text.

[USR] <TRIP_INFO> {trip_info_1} </TRIP_INFO> <TRANSPORT_OPTIONS> {transport_options_1} </TRANSPORT_OPTIONS> <CHOICE> {choice_1} </CHOICE> <TRIP_INFO> {trip_info_2} </TRIP_INFO> <TRANSPORT_OPTIONS> {transport_options_2} </TRANSPORT_OPTIONS> <CHOICE> {choice_2} </CHOICE> <TRIP_INFO> {trip_info_3} </TRIP_INFO> <TRANSPORT_OPTIONS> {transport_options_3} </TRANSPORT_OPTIONS> <CHOICE> {choice_3} </CHOICE> <TRIP_INFO> {trip_info_4} </TRIP_INFO> <TRANSPORT_OPTIONS> {transport_options_4} </TRANSPORT_OPTIONS> <CHOICE> {choice_4} </CHOICE> <TRIP_INFO> {trip_info_5} </TRIP_INFO> <TRANSPORT_OPTIONS> {transport_options_5} </TRANSPORT_OPTIONS> <CHOICE> {choice_5} </CHOICE> <TRIP_INFO> {trip_info_6} </TRIP_INFO> <TRANSPORT_OPTIONS> {transport_options_6} </TRANSPORT_OPTIONS> <CHOICE> Please predict the travel mode for this trip. </CHOICE>

---

**Prompt B.4:** *Swissmetro* **- TextGrad**

[INITIAL FULL PROMPT + SOLUTION] Task: Estimate the probability distribution over three travel modes (Swissmetro, Train, Car) for a single trip.
<TRIP_INFO> {trip_info} </TRIP_INFO>
<TRANSPORT_OPTIONS> {transport_options} </TRANSPORT_OPTIONS>
Solution (JSON): {"Swissmetro": 0.333, "Train": 0.333, "Car": 0.334}
[GRADING PROMPT] You are a transport-economics expert. Given the trip info, transport options, and predicted probabilities in the user's message, output a single line ONLY: Score: <float between 0 and 1> 1 = probabilities look highly reasonable, 0 = implausible. Remember: THE PREDICTION MUST BE A JSON DICT.

# C Additional Experiments: Reasoning LLMs and End-to-End Baselines

**Purpose and setup.** This section probes how much backbone model capacity matters on our tasks. For a controlled comparison, we randomly sample 100 individuals from the 500-person pool to form a compact evaluation subset (same preprocessing, metrics, and decoding settings as in the main experiments). For each individual, we randomly sample one record. We evaluate ATHENA with five backbones: two state-of-the-art open-source reasoning models (Qwen3-32B, DeepSeek-R1-Distill-Qwen-32B) and three leading commercial offerings (GPT-4o-mini, GPT-4o, Gemini-2.0-Flash). Across both tasks, ATHENA attains *state-of-the-art classification performance* among LLM-based methods—consistently delivering the highest *Accuracy* and *F1*, with *AUC* that is competitive or superior to prompt-only LLM baselines (see Tables 13 and 1).

**Structure dominates model size; stronger reasoning yields modest, consistent gains.** Under ATHENA, swapping GPT-4o-mini for larger "reasoning" backbones (e.g., GPT-4o, Qwen3-32B, DeepSeek-R1) yields *incremental* but *consistent* improvements, especially on the more interaction-heavy *Vaccine* task. The effect is smaller on *Swissmetro*, where dominant explanatory factors (time/cost) are already well captured by the *symbolic discovery → textual refinement* pipeline. Intuitively, Stage 1 constrains the hypothesis space to interpretable utility forms, and Stage 2 makes small, directed edits to those forms; this turns the problem into guided search plus local adjustments. As a result, *structural bias* (symbolic utility discovery + semantic adaptation) shoulders most of the lift, while *backbone capacity* primarily fine-tunes edge cases (nonlinear interactions, atypical profiles), producing a steady but not dramatic gain.

**Prompt-only methods are brittle and poorly calibrated; ATHENA regularizes both decisions and probabilities.** Zero-shot / CoT / Few-shot prompting shows visible volatility across metrics: *Accuracy/F1* can spike on one dataset yet drop on another, and *AUC/CE* often swing with decoding details (temperature, sampling count, score-to-probability mapping). ATHENA markedly reduces this variance: the symbolic stage enforces cross-person consistency (shared operators, shared concept library), while the textual refinement stage adjusts *within* those constraints, leading to better class separability and more conservative probability mass. Empirically this manifests as stronger and more stable *F1/AUC*, with *CE* reflecting improved calibration compared to prompt-only baselines. In short, structure acts as *regularization* for both decisions and confidence.

**End-to-end baselines trail on interpretability and robustness; ATHENA's decomposition captures heterogeneity with explicit utility logic.** Machine learning-based models can be competitive on single metrics in isolated settings, but they do not expose explicit, policy-relevant utility functions and are less consistent across tasks/splits. They must implicitly learn both *which* attributes matter and *how* they combine, from scratch. ATHENA instead *decouples* the problem: Stage 1 discovers globally interpretable utility structure (operators, interactions), and Stage 2 adapts those structures to individual semantics. This yields *(i)* stronger across-task consistency in *Accuracy/F1/AUC*, *(ii)* end-to-end interpretability of the discovered utilities.

Table 13: Performance comparison across methods on *Swissmetro* and *Vaccine* datasets.

| Method | LLM Model | Swissmetro | | | | Vaccine | | | |
|---|---|---|---|---|---|---|---|---|---|
| | | Acc.↑ | F1↑ | CE↓ | AUC↑ | Acc.↑ | F1↑ | CE↓ | AUC↑ |
| Zeroshot | gemini-2.0-flash | 0.5800 | 0.3046 | 0.9059 | 0.6829 | 0.6000 | 0.5386 | 0.8317 | 0.7433 |
| | GPT-4o-mini | 0.6100 | 0.2763 | 0.9253 | 0.5556 | 0.5500 | 0.5302 | 0.8271 | 0.7500 |
| | GPT-4o | 0.5900 | 0.3310 | 0.8646 | 0.6946 | 0.6000 | 0.5465 | 0.8052 | 0.7306 |
| | Qwen3 | 0.5900 | 0.4158 | 1.4047 | 0.6126 | 0.5400 | 0.5729 | 0.8676 | 0.7519 |
| | DeepSeek-r1 | 0.5900 | 0.4339 | 1.2473 | 0.6608 | 0.6300 | **0.6531** | 0.8244 | 0.7692 |
| Zeroshot-CoT | gemini-2.0-flash | 0.5300 | 0.2809 | 0.9858 | 0.6409 | 0.6100 | 0.5485 | 0.8128 | 0.7604 |
| | GPT-4o-mini | 0.6100 | 0.2763 | 0.9109 | 0.6156 | 0.5900 | 0.5820 | 0.8161 | 0.7714 |
| | GPT-4o | 0.5800 | 0.3162 | 0.9237 | 0.6404 | 0.6300 | 0.5717 | 0.7785 | 0.7700 |
| | Qwen3 | 0.6200 | 0.4632 | 1.7101 | 0.6284 | 0.5400 | 0.5843 | 0.9073 | 0.7364 |
| | DeepSeek-r1 | 0.5800 | 0.3018 | 0.9522 | 0.6173 | 0.5200 | 0.5492 | 0.8761 | 0.7373 |
| Few-shot | gemini-2.0-flash | 0.7200 | 0.6922 | 10.0922 | 0.7984 | 0.5300 | 0.5508 | 14.2531 | 0.6747 |
| | GPT-4o-mini | 0.6800 | 0.5320 | 5.0402 | 0.7516 | 0.5200 | 0.5278 | 7.7066 | 0.6975 |
| | GPT-4o | 0.7300 | 0.6654 | 3.9386 | 0.8423 | 0.5700 | 0.5967 | 5.6261 | 0.7467 |
| | Qwen3 | 0.7400 | 0.6760 | 7.0435 | 0.8033 | 0.5400 | 0.5487 | 9.5013 | 0.7060 |
| | DeepSeek-r1 | 0.7000 | 0.6282 | 3.6154 | 0.8439 | 0.5500 | 0.5392 | 8.8895 | 0.6855 |
| TextGrad | gemini-2.0-flash | 0.5400 | 0.2432 | 1.1934 | 0.4718 | 0.5000 | 0.4511 | 4.1290 | 0.7345 |
| | GPT-4o-mini | 0.5700 | 0.3111 | 0.9551 | 0.5292 | 0.5000 | 0.4686 | 4.7960 | 0.6460 |
| | GPT-4o | 0.5600 | 0.3316 | 0.9441 | 0.6256 | 0.5600 | 0.5321 | 2.3468 | 0.6721 |
| | Qwen3 | 0.5100 | 0.3669 | 2.0180 | 0.5322 | 0.4600 | 0.4276 | 6.4994 | 0.6303 |
| | DeepSeek-r1 | 0.5800 | 0.3356 | 0.9631 | 0.6344 | 0.4300 | 0.4235 | 3.1080 | 0.5948 |
| ATHENA (ours) | gemini-2.0-flash | **0.7900** | 0.7185 | **0.6121** | **0.9153** | 0.6500 | 0.5978 | 0.8305 | 0.7998 |
| | GPT-4o-mini | 0.7600 | **0.7304** | 1.4208 | 0.8577 | 0.6500 | 0.6079 | 0.8034 | 0.8133 |
| | GPT-4o | 0.7700 | 0.7085 | 1.0417 | 0.8697 | **0.6700** | 0.6213 | **0.7765** | **0.8279** |
| | Qwen3 | 0.7400 | 0.7040 | 4.9132 | 0.7754 | 0.5700 | 0.5650 | 1.1393 | 0.7637 |
| | DeepSeek-r1 | 0.7100 | 0.6612 | 0.8437 | 0.8353 | 0.6600 | 0.6501 | 0.8115 | 0.8212 |

# D  Empirical Scalability Evidence

We benchmarked wall-clock time and token usage using `gpt-4o-mini` on the Swissmetro subset.

## D.1  Stage 2 – Individual adaptation

Table 14: Runtime and token usage for Stage 2 (individual-level semantic adaptation) under different iteration counts $T'$.

| $T'$ | Time (s) | s/iter | tokens/iter |
|---|---|---|---|
| 1 | 48.16 | 48.16 | 1079.6 |
| 3 | 176.89 | 58.96 | 1195.83 |
| 5 | 315.66 | 63.13 | 1249.28 |

## D.2  Stage 1 – Group-level discovery

Table 15: Runtime and token usage for Stage 1 (group-level symbolic utility discovery) under different iteration counts $T$.

| $T$ | Time (min) | tokens total |
|---|---|---|
| 5 | 30.61 | 215,281 |
| 15 | 36.69 | 251,974 |
| 30 | 65.64 | 479,751 |

These results confirm that runtime and token usage scale approximately linearly with the number of iterations, consistent with the theoretical analysis.

# E Prompts

Take **Swissmetro** dataset as an example.

---

**Prompt E.1:** *Swissmetro* - **Symbolic Utility Initialization**

Step 1:
[SYS] You are a transportation planner specializing in analyzing the relationships among various factors that influence travel behavior. You will be provided with two types of information: individual features (delimited by <FEATURES> and </FEATURES>) and preliminary travel mode knowledge (delimited by <KNOWLEDGE> and </KNOWLEDGE>). Your task is to carefully review these inputs and in detailed sentence describe how the provided features interrelate. Ensure your response includes as many specific details as possible about the relationships, but do not propose any new features or suggest modifications to the existing ones. Example: Time: quadratic, Cost: log, luggage: linear.
[USR] <GROUP DESCRIPTION>{description}</GROUP DESCRIPTION> <FEATURES>{features}</FEATURES> <KNOWLEDGE>{knowledge}</KNOWLEDGE>
You should ONLY provide the relations between the features. YOU MUST return your assumption in this exact format: "'["relation_0","relation_1", ...]
Step 2:
[SYS] You are a helpful assistant that proposes mathematical expressions based on some provided suggestions. Your goal is to:
0. **Task**: Generate utility functions for travel mode choice of group of {description}.
1. **Use only** the specified variables: {variables}
2. **Represent all constants** with the symbol "C", and all coefficients with the symbol "K".
3. **Restrict** yourself to the following operators: operators
4. **For each group**, suggest utility functions for train, car, and Swissmetro respectively. Your response must: - Propose exactly **{N}** groups of expressions. - MUST return in this exact format: "'[("expressions_car","expressions_train","expressions_metro"), ...]"', replace expressions_mode with your proposed expressions.
[USR] Suggestions: {suggestions}

---

**Prompt E.2:** *Swissmetro* - **Results Analysis**

[SYS] You are a creative and insightful mathematical research assistant. You have been provided with two sets of utility expressions: one function group labeled "Good Expressions" and one labeled "Bad Expressions." Your objective is to hypothesize about the underlying assumptions or principles that might generate the good expressions yet exclude the bad ones.
Key Points:
1. Focus primarily on the good expressions' mathematical structures and any connections they might have to physical or applied contexts.
2. Capital "C" in any expression is just an arbitrary constant.
3. Do not discuss or compare the expressions in terms of their simplicity or complexity.
4. Provide your reasoning step by step, but keep it very concise and genuinely insightful. No more than 5 lines.
[USR] Good Expression 1: (train: {texpr1}, car: {cexpr1}, metro: {mexpr1}), accuracy: {acc1}
Good Expression 2: (train: {texpr2}, car: {cexpr2}, metro: {mexpr2}), accuracy: {acc2}
Bad Expression 1: (train: {bexpr1}, car: {bexpr2}, metro: {bexpr3}), accuracy: {acc3}
Above expressions are travel mode choice utility functions of group of {description}. Propose {N} hypotheses that would be appropriate given the expressions. Provide short commentary for each of your decisions. Do not talk about topics related to the simplicity or complexity of the expressions. I want ideas that are unique and interesting enough to amaze the world's best mathematicians.

**Prompt E.3:** *Swissmetro* **- Crossover**

[SYS] You are a helpful assistant that recombines two mathematical expressions based on some provided suggestions. Your goal is to produce new expressions that:
1. Blend or merge elements from both reference expressions in a way that reflects the suggestions.
2. Adhere to the following constraints:
- You may only use the variables in library: {variables}
- All constants must be represented with the symbol C
- Only the following operators are allowed: {operators}
Guidelines:
- Propose exactly {N} new expressions.
- Each new expression should integrate elements of both reference expressions. You can also propose new terms with variables that are in the library but not in the old expressions.
- If any suggestions appear contradictory, reconcile them reasonably.
MUST return in this exact format:
```
[("expressions_car","expressions_train","expressions_metro"), ...]
```
Replace `expressions_` with your proposed expressions.
[USR] Suggestion: {suggestions}
Reference Expression group 1: (train: {texpr1}, car: {cexpr1}, metro: {mexpr1})
Reference Expression group 2: (train: {texpr2}, car: {cexpr2}, metro: {mexpr2})

Propose {N} expressions that would be appropriate given the suggestions and references.

---

**Prompt E.4:** *Swissmetro* **- Mutation**

[SYS] You are a helpful assistant that generates mutated variants of a **triplet** of mathematical expressions (car, train, metro) based on provided mutation strategies. Your goal is to produce new expression triplets that: 1. Mutate the reference expressions by applying mutation operations (e.g., adjust coefficients, swap variables, alter operators) in a way that reflects the suggestions. 2. Adhere to the following constraints: - You may only use the variables in library: {variables} - All constants must be represented with the symbol C - Only the following operators are allowed: {operators}
Guidelines: - Produce exactly {M} mutated **triplets**. - Within each triplet you must provide one mutated expression for **car**, one for **train** and one for **metro**. - A mutation can modify any combination of variable, operator or constant, but each expression must remain syntactically valid under the constraints.
MUST return in this exact format:
```
[("mut_car1","mut_train1","mut_metro1"), ...]
```
[USR] Generate {M} mutated variants of the following mathematical expression triplet according to these mutation strategies: – You may only use variables from: {variables} – All constants must be written as C – Only these operators are allowed: {operators}
Mutation strategies: {suggestions}
Reference expressions: (car): {cexpr} (train): {texpr} (metro): {mexpr}
Please return exactly {M} new, syntactically valid triplets in the JSON list format shown above.

**Prompt E.5:** *Swissmetro* **- Semantic Adaptation Initialization**

[SYS] You are a travel-behavior preference selector. You will be given two blocks of information:
<DEMOGRAPHICS> ... </DEMOGRAPHICS> <UTILITY_FUNCTION> ... </UTIL-ITY_FUNCTION>
Your goal: choose the single best-matching high-level preference template for this group **exactly** from the catalogue below and output **only** the template name (uppercase).
CATALOGUE - TIME_EFFICIENCY : travellers primarily minimise total travel time.
- COST_SAVING : travellers primarily minimise direct monetary cost.
- COMFORT_SEEKING : travellers value comfort/service frequency and dislike crowding.
- BALANCED : sensitivities are evenly distributed across factors.
...//OTHER POSSIBLE TEMPLATE
Return nothing else — no commentary, no punctuation, just the template name.
[USR] <DEMOGRAPHICS>{demographics}</DEMOGRAPHICS>
<UTILITY_FUNCTION>{utility}</UTILITY_FUNCTION>

---

**Prompt E.6:** *Swissmetro* **- Semantic Adaptation Loss Function**

Evaluate the travel mode prediction based on the individual's profile and alternatives. Compare it to the actual choice and identify any discrepancies. Be concise and focus on why the prediction might be incorrect. Return 0 if they match, 1 otherwise.

---

**Prompt E.7:** *Swissmetro* **- Prediction**

[SYS] You are a decision assistant that recommends the most suitable travel mode for an individual trip by estimating a probability distribution over three options: Swissmetro, Train, and Car.
You will receive three blocks: <TEMPLATE> ... optimized preference template ... </TEM-PLATE>
<PROFILE> ... individual profile ... </PROFILE>
<ALTERNATIVES> ... attributes of Swissmetro, Train, and Car ... </ALTERNATIVES>
**Instructions:** 1. **Use the <TEMPLATE> as a guide** for understanding the individual's likely preference bias (e.g., time efficiency, cost saving, comfort seeking, balanced).
2. **Analyze the <PROFILE>** (age, gender, income, trip details) **and the <ALTERNA-TIVES>** (travel time, cost, headway).
3. **Estimate and output a probability** for each travel mode, such that all three probabilities sum to 1.
**Output format (JSON only):** "'json { "Swissmetro": <float between 0 and 1>, "Train": <float between 0 and 1>, "Car": <float between 0 and 1> } "' No additional text; just the JSON object with normalized probabilities.
[USR] <TEMPLATE>{template_name}</TEMPLATE> <PROFILE> {individual_block} </PROFILE> <ALTERNATIVES> {options} </ALTERNATIVES>

