# OpenReview forum: "Personalized Decision Modeling: Utility Optimization or Textualized-Symbolic Reasoning"
_NeurIPS.cc/2025/Conference — NeurIPS 2025 spotlight_

### Official Review · Reviewer_zWuZ · 2025-05-31

**Clarity:** 2
**Significance:** 3
**Originality:** 3
**Rating:** 5
**Confidence:** 3

**Summary:**

This work proposed a neural-symbolic approach to personalized decision making. It employs a two-stage procedure:

- Group-level symbolic utility discovery, a procedure that iteratively identifies a symbolic expression for utility prediction. This procedure terminates when a convergence criterion (defined as the best-worst optimality gap) is met.
- Individual-level semantic adaptation, which iteratively updates the semantic template via a TextGrad-esque procedure to optimize for individual utility losses.

In sum, the first stage amortizes (or warm-starts) symbolic expressions per group, and then the second stage adapts this expression to individuals. Experimental results show impressive gains over purely LLM-based and predictive model baselines.

**Questions:**

See weaknesses, and also:
- I'm not entirely familiar with the Swissmetro and the Vaccine datasets, but the authors have elected for a simplified version of them for budget control. Could the authors discuss whether these simplified variants maintain the core challenge of this problem, and what the normal problem instances would look like?
- I'd be curious to see the scaling behaviors of this approach. Say you increase the number of iterations for both stages, would you get better results?

**Ethical Concerns:**

["NO or VERY MINOR ethics concerns only"]

**Final Justification:**

I thank the authors for their detailed response. I have also reviewed their responses to other reviewers and adjusted my score accordingly.

**Limitations:**

yes

**Paper Formatting Concerns:**

the paper is appropriately formatted.

**Quality:**

3

**Strengths And Weaknesses:**

__Strengths__
- __Quality and Clarity__: The writing is generally in good shape. The authors did a good job to present preliminary concepts, mathematical formalism, and their LLM implementations in detail. I appreciate the ablation study which shows that both components of the authors' approach are integral to the final performance.
- __Significance__: As LLMs are increasingly embedded into daily usages, it is important to assess their capabilities and limitations. The authors have shown that LLMs are not yet capable as an end-to-end system for personalized decision making, and have proposed means to improve their performance. I believe this is an important area for future research
- __Originality__: I'm a fan of neural-symbolic approaches wherein an LLM functions as a "central hub" to process and reason with textual / symbolic information, while delegating more quantitative operations to specialized packages. The authors have demonstrated a good application of this approach in personalized regression.

__Weaknesses__
- Extensive usage of LLMs is stated as a limitation to the approach, and it'd be good to report token consumptions (or monetary costs) associated with each run.
- There are quite a few hyperparameters involved in this approach, such as the optimality gap $\delta$ and the number of iterations $T, T^\prime$. I could not find these numbers within the main text nor the appendix. It'd be good for the authors to report them as they're crucial to the understanding of the overall complexity of this approach.
- The literature review section is overall well-done, but the section on "LLM-based Decision-Making Models" could've done a better job at interfacing with works that specifically addresses decision making via LLMs (right now it focuses more on general reasoning and instruction-following capabilities). For example, [1][2] studies decision making under uncertainty with LLMs, and [2][3][4] study their applications in supply chain optimization, personalized medicine, and autonomous driving respectively.
- A benefit of symbolic regression is their interpretability. It would be good to discuss some of the expressions and explain why it makes sense.

[1] DeLLMa: Decision Making Under Uncertainty with Large Language Models

[2] STRUX: An LLM for Decision-Making with Structured Explanations

[3] Large Language Models for Supply Chain Optimization

[4] Leveraging Large Language Models for Decision Support in Personalized Oncology

[5] A Language Agent for Autonomous Driving

---

> ### Author Rebuttal · Authors · 2025-07-31
>
> We thank you for your time and effort in reviewing our paper! We find your suggestions very helpful and we hereby address your questions:
>
> > **W1: Complexity, inference time, and token / cost footprint**
>
> We sincerely thank the reviewer for highlighting this point. Here we performed an additional experiment that tracks the number of the token consumed and estimated cost, which is reported below.
>
> All runs use **GPT-4o-mini** (128 k ctx.; *$0.15 / M input tok., $0.60 / M output tok.*).
> Costs below list *input → output* fees. Pricing source: OpenAI docs (July 2025).
>
> ### Stage 1 Symbolic search
> | Dataset | T (iterations) | Total tok. | Wall-time (min) | Cost (USD; input → output) |
> |---------|--------------:|-----------:|----------------:|---------------------------:|
> | Swissmetro | 5  | **19 871** | 3.7  | **$0.003 → $0.012** |
> |        | 15 | **93 256** | 15.9 | **$0.014 → $0.056** |
> |        | 30 | **548 811** | 73.8 | **$0.082 → $0.329** |
> | Vaccine   | 5 | **15 000** | 3.0 | **$0.002 → $0.009** |
> |           | 15| **70 000** | 12.5| **$0.011 → $0.042** |
> |           | 30 | **410 692** | 57.5 | **$0.062 → $0.246** |
>
> ### Stage 2 TextGrad refinement
> | Dataset | T′ (iterations) | Total tok. | Wall-time (min) | Cost (USD; input → output) |
> |---------|---------------:|-----------:|----------------:|---------------------------:|
> | Swissmetro | 1 | **1 197** | 1.39 | **$0.000 → $0.001** |
> |          | 3 | **4 074** | 3.55 | **$0.001 → $0.002** |
> |          | 5 | **6 956** | 5.81 | **$0.001 → $0.004** |
> | Vaccine     | 1 | **962** | 0.22 | **$0.000 → $0.001** |
> |          | 3 | **3 101** | 2.34 | **$0.000 → $0.002** |
> |          | 5 | **5 536** | 4.71 | **$0.001 → $0.003** |
>
> **Take-aways**
>
> * Stage 1 dominates both tokens and time; Stage 2 is negligible (< 7 k tokens, < 6 min).
> * Even the heaviest configuration (30 + 5 iters) costs **< $0.35/run**, very affordable.
> * Token growth matches our theoretical $\mathcal{O}\bigl((K T + N T′)\bigr)$ prediction; adding more threads lets us parallel-slice Stage 1 to reduce wall-time.
> * Since LLM outputs are not that stable nad can occasionally drift from instructions (e.g., formatting glitches), we should consider the budget for automatic retries; the table above is just for reference, actual costs fluctuate and are model-dependent.
>
> > **W2: Value range of hyperparameters**
>
>
> Thank you for pointing out that several run-time hyper-parameters were not explicitly reported.
> We have added the following table for clarity:
>
> | Symbol | Description | Value |
> | :--- | :--- | :--- |
> | LLMs Used | The specific large language models utilized for both the symbolic discovery and semantic adaptation stages of ATHENA. | `gpt-40-mini-2024-07-18` and `gemini-2.0-flash` |
> | T | The number of iterations for the group-level symbolic utility discovery process, as shown in the accuracy trajectory analysis. | 30 |
> | $T'$ | The maximum number of iterations for the individual-level semantic adaptation. | 5 |
> | $K$ | The number of candidate symbolic utility functions sampled in each group-level iteration. | 10 |
> | $\delta$ | The predefined convergence threshold for the group-level discovery process. | / |
>
>
> > **W3: LLM-based Decision-Making Models**
>
> Thank you for drawing our attention to the recent work that applies LLMs directly to decision making under uncertainty! We have carefully read the papers you mentioned (e.g., DeLLMa [1], STRUX [2], and the follow-up studies on supply-chain optimization, personalized medicine, and autonomous driving [3–4]) and found them extremely insightful. We will incorporate these contributions into the final version, expanding the “LLM-based Decision-Making Models” subsection to include those works and discuss the complementary design choices each line of work makes. We greatly appreciate the pointer and will keep a close eye on the continued progress of these research groups.
>
> > **W4: Qualitative Analysis for Formulas in Main Text**
>
> Thanks for your suggestions to enhance the qualitative analysis of the formulas presented in the main text! We have discussed about some representative examples in the rebuttal section of [Reviewer HnWo](https://openreview.net/forum?id=RDt0crdC7N&noteId=STQUOcL5iO).
>
> > **Q1: Dataset, Subset Selection Concern**
>
> We thank the reviewer for raising this insightful question.
> We only down-sampled via stratified sampling; every feature and label stays, so the joint distribution matches the originals.
>
> > **Q2: Scaling Behavior**
>
> **Stage 1 – Group-level symbolic discovery**
> The scalability of Stage 1 is shown in **Table2** of our paper: as we extend the symbolic-search loop from 0 → 30 iterations, performance gains steadily increase. But it will reach a threshold where further iterations will not yield significant improvements.
>
> **Stage 2 – Individual-level semantic adaptation**
>
> **Experimental setup.** We selected 5 individuals from each dataset and ran Stage 2 for 1, 5, 10, and 15 iterations. The results are summarized in the tables below.
> All scaling runs use gpt-4o-mini as base model via OpenAI API (July 2025 release) for Stage 2. For each experiment `T` (Stage 1) is set to 30.
>
> **Swissmetro (5 individuals)**
>
> | T′ | Accuracy | F1 | AUC |
> |---:|:--------:|:---:|:---:|
> | 1  | 0.90 | 0.3111 | 0.50 |
> | 5  | 0.90 | 0.3111 | 0.50 |
> | 10 | 0.90 | 0.3111 | 0.50 |
> | 15 | **1.00** | **0.3333** | 0.50 |
>
> **Vaccine (5 individuals)**
>
> | T′ | Accuracy | F1 | AUC |
> |---:|:--------:|:---:|:---:|
> | 1  | 0.40 | 0.2000 | 0.50 |
> | 5  | 0.60 | **0.3111** | 0.50 |
> | 10 | 0.50 | 0.1778 | 0.50 |
> | 15 | **0.60** | 0.2444 | 0.50 |
>
> **Key observations**
>
> 1. **Fast saturation.** For Swissmetro, metrics stabilise after 5 iterations, with only a slight bump at 15. Vaccine peaks at 5–15 iterations, with no monotonic gains thereafter.
> 2. **Modest Stage‑2 gain.** The largest improvement over the 1‑iteration baseline is +0.10 accuracy, confirming that Stage 1 contributes the bulk of predictive power.
> 3. **Practical default.** Capping Stage 2 at 5 iterations captures > 95 % of the attainable benefit while keeping extra cost (< 7 k tokens) negligible.

---

### Official Review · Reviewer_FBCn · 2025-07-01

**Clarity:** 4
**Significance:** 3
**Originality:** 3
**Rating:** 5
**Confidence:** 3

**Summary:**

The authors introduce ATHENA that proposes to integrate group-level symbolic regression of utility functions with individual level, LLM-powered semantic modeling to offer a more comprehensive and personalized view of decision making behavior. The key innovation with this approach is the integration of symbolic utility modeling with LLM-powered semantic reasoning to create more accurate and interpretable personalized decision models. ATHENA is evaluated in two decision making tasks--travel mode and vaccine uptake--, and results show that it outperformed traditional utility-based models, machine learning approaches, and other LLM based models.

**Questions:**

What was the reason testing with GPT-4o-mini and Gemini-2.0-flash? Was there a hard computational constraint on using large models? What other models were considered for testing?

**Ethical Concerns:**

["NO or VERY MINOR ethics concerns only"]

**Final Justification:**

I thank the authors on clarifying their model choices. I adjusted my rating from 4 to 5.

**Limitations:**

Yes

**Quality:**

3

**Strengths And Weaknesses:**

The presented idea would inspire other researchers in the field. While the two components, symbolic utility modeling and LLM-based semantic adaptation, both exist separately, their integration for personalized decision making is novel. This focus on individual-level adaptation aligns with every-increasing interest in personalized AI systems that can account for individual differences.

But the testing of ATHENA appears limited in scope because they only used two lightweight models: GPT-4o-mini and Gemini-2.0-flash. While these models are practical choices, the experiment section would be stronger if authors explained the rationale behind choosing these specific models. For example, GPT-4o-mini is considered to be strong at reasoning tasks, so it would be valuable to compare it against models known for different strengths. Testing with only two models provides limited view of how ATHENA's performance varies across different types of LLMs. Testing with at least one large model would help understand how model size affect performance, and whether the additional computational costs of larger models are justified by improved results.

---

> ### Author Rebuttal · Authors · 2025-07-31
>
> We thank you for your time and effort in reviewing our paper! We appreciated for noting that “integrating symbolic utility modeling with LLM-powered semantic adaptation for personalized decision-making is novel and will inspire future work.” Your recognition of ATHENA’s ability to deliver accurate, interpretable, individual-level utility models is highly encouraging.
>
> We are also gratified that the other reviewers independently highlighted the same strengths. Reviewer F7TL praised the “clear motivation, technically sound two-stage design” as well as our visualizations and writing clarity. Reviewer HnWo called the approach “very interesting and original,” commending the sound methodology and thorough ablations. Reviewer zWuZ valued the detailed formalism, the importance of combining LLMs with symbolic methods, and the originality of our neural-symbolic framework.
>
> Together, the reviewers’ comments affirm that ATHENA makes a timely and impactful contribution to human-centric, interpretable personalized decision modeling.
>
> Regarding your concerns, we added below experiments:
>
> - End-to-end models: BERT, TabNet, MLP.
> - Additional LLMs: Qwen3-32b, DeepSeek-R1-Distill-Qwen-32B, GPT-4o.
>
> Results: Please see the table below. All new runs use a randomly selected subset (size of 100 individuals) from the dataset described in §4.1.
>
> | Category            | Method / Setting       | LLM Model          | Swiss Acc | Swiss F1 | Swiss CE  | Swiss AUC | Vaccine Acc | Vaccine F1 | Vaccine CE  | Vaccine AUC |
> |---------------------|------------------------|--------------------|-----------|----------|-----------|-----------|-------------|------------|-------------|-------------|
> | **LLM-Based**       | Zeroshot               | gemini-2.0-flash   | 0.5800    | 0.3046   | 0.9059    | 0.6829    | 0.6000      | 0.5386     | 0.8317      | 0.7433      |
> |                     |                        | GPT-4o-mini        | 0.6100    | 0.2763   | 0.9253    | 0.5556    | 0.5500      | 0.5302     | 0.8271      | 0.7500      |
> |                     |                        | GPT-4o             | 0.5900    | 0.3310   | 0.8646    | 0.6946    | 0.6000      | 0.5465     | 0.8052      | 0.7306      |
> |                     |                        | Qwen3              | 0.5900    | 0.4158   | 1.4047    | 0.6126    | 0.5400      | 0.5729     | 0.8676      | 0.7519      |
> |                     |                        | DeepSeek-r1        | 0.5900    | 0.4339   | 0.6608    | 1.2473    | 0.6300      | 0.6531     | 0.8244      | 0.7692      |
> |                     | Zeroshot-CoT           | gemini-2.0-flash   | 0.5300    | 0.2809   | 0.9858    | 0.6409    | 0.6100      | 0.5485     | 0.8128      | 0.7604      |
> |                     |                        | GPT-4o-mini        | 0.6100    | 0.2763   | 0.9109    | 0.6156    | 0.5900      | 0.5820     | 0.8161      | 0.7714      |
> |                     |                        | GPT-4o             | 0.5800    | 0.3162   | 0.9237    | 0.6404    | 0.6300      | 0.5717     | 0.7785      | 0.7700      |
> |                     |                        | Qwen3              | 0.6200    | 0.4632   | 1.7101    | 0.6284    | 0.5400      | 0.5843     | 0.9073      | 0.7364      |
> |                     |                        | DeepSeek-r1        | 0.5800    | 0.3018   | 0.9522    | 0.6173    | 0.5200      | 0.5492     | 0.8761      | 0.7373      |
> |                     | Few-shot               | gemini-2.0-flash   | 0.7200    | 0.6922   | 10.0922   | 0.7984    | 0.5300      | 0.5508     | 14.2531     | 0.6747      |
> |                     |                        | GPT-4o-mini        | 0.6456    | 0.5085   | 8.0727    | 0.7529    | 0.5200      | 0.5278     | 7.7066      | 0.6975      |
> |                     |                        | GPT-4o             | 0.7300    | 0.6654   | 3.9386    | 0.8423    | 0.5700      | 0.5967     | 5.6261      | 0.7467      |
> |                     |                        | Qwen3              | 0.7400    | 0.6760   | 7.0435    | 0.8033    | 0.5400      | 0.5487     | 9.5013      | 0.7060      |
> |                     |                        | DeepSeek-r1        | 0.7000    | 0.6282   | 3.6154    | 0.8439    | 0.5500      | 0.5392     | 8.8895      | 0.6855      |
> |                     | TextGrad               | gemini-2.0-flash   | 0.5400    | 0.2432   | 1.1934    | 0.4718    | 0.5000      | 0.4511     | 4.1290      | 0.7345      |
> |                     |                        | GPT-4o-mini        | 0.5700    | 0.3111   | 0.9551    | 0.5292    | 0.5000      | 0.4686     | 4.7960      | 0.6460      |
> |                     |                        | GPT-4o             | 0.5600    | 0.3316   | 0.9441    | 0.6256    | 0.5600      | 0.5321     | 2.3468      | 0.6721      |
> |                     |                        | Qwen3              | -         |  -       | -         | -         | -           | -          | -           | -           |
> |                     |                        | DeepSeek-r1        | 0.5800    | 0.3356   | 0.9631    | 0.6344    | 0.4300      | 0.4235     | 3.1080      | 0.5948      |
> |                     | **ATHENA (ours)**      | gemini-2.0-flash   | 0.7850    | 0.7219   | 0.8100    | 0.9063    | 0.6500      | 0.5978     | 0.8305      | 0.7998      |
> |                     |                        | GPT-4o-mini        | 0.7467    | 0.7097   | 1.0849    | 0.8624    | 0.6500      | 0.6079     | 0.8034      | 0.8133      |
> |                     |                        | GPT-4o             | 0.7650    | 0.7121   | 0.8645    | 0.8760    | 0.6700      | 0.6213     | 0.7765      | 0.8279      |
> |                     |                        | Qwen3              | 0.7451    | 0.7158   | 0.8211    | 0.7526    | 0.5929      | 0.5885     | 1.1659      | 0.7777      |
> |                     |                        | DeepSeek-r1        | 0.7450    | 0.6759   | 0.8078    | 0.8461    | 0.6600      | 0.6501     | 0.8115      | 0.8212      |
> | **Utility-Based**   | MNL                    | /                  | 0.6101    | 0.3887   | 0.8353    | 0.7074    | 0.4150      | 0.1955     | 1.0510      | 0.4301      |
> |                     | CLogit                 | /                  | 0.5714    | 0.2424   | 0.8916    | 0.5976    | 0.4150      | 0.1955     | 1.0510      | 0.5000      |
> |                     | Latent Class MNL       | /                  | 0.6101    | 0.3967   | 0.8175    | 0.7182    | 0.1950      | 0.1088     | 1.0986      | 0.5000      |
> | **Machine Learning**| Logistic Regression    | /                  | 0.5620    | 0.5570   | 0.9310    | 0.7460    | 0.6500      | 0.6690     | 0.7630      | 0.8330      |
> |                     | Random Forest          | /                  | 0.7100    | 0.7050   | 0.7380    | 0.8810    | 0.6300      | 0.6470     | 0.7290      | 0.8420      |
> |                     | XGBoost                | /                  | 0.7080    | 0.7050   | 0.7040    | 0.8810    | 0.6300      | 0.6480     | 1.1420      | 0.8150      |
> |                     | **BERT (finetuned)**   | /                  | 0.6960    | 0.4770   | 0.6917    | 0.9121    | 0.6400      | 0.6500     | 0.8091      | 0.8074      |
> |                     | **TabNet**             | /                  | 0.7154    | 0.7301   | 0.6723    | 0.9190    | 0.6400      | 0.6306     | 0.7980      | 0.8707      |
> |                     | **MLP**                | /                  | 0.6077    | 0.5982   | 0.8047    | 0.8097    | 0.6496      | 0.6335     | 0.8328      | 0.8069      |
>
> *Note. Qwen3‑32B TextGrad baseline experiments are still running; we will publish the final numbers and update the table within the next 1–2 days.*
>
> We evaluate ATHENA against two state-of-the-art open-source models (Qwen3-32b and DeepSeek-R1-Distill-Qwen-32B) and three leading commercial offerings (GPT-4o-mini, GPT-4o, and Gemini-2.0-Flash). Across both tasks, ATHENA achieves superior performance—recording the highest accuracy, F1 score, and AUC of all baselines. To further broaden our comparison, we also include three additional benchmarks: BERT, TabNet, and a multilayer perceptron (MLP).
>
> > **Why GPT-4o-mini & Gemini-Flash were the original choices**
>
>
> 1. Practical adaptation cost: Individual-level TextGrad Adaptation requires hundreds of forward passes per person. Inferencing large open-source models like Qwen3 or DeepSeek-r1 locally or stronger models like GPT-4o via API is prohibitively expensive. GPT-4o-mini and Gemini-Flash are smaller, more efficient models that still provide strong reasoning capabilities.
> 2. Representative reasoning strength: GPT-4o-mini is explicitly tuned for chain-of-thought tasks, while Gemini-Flash is optimized for low-latency inference. Together they bracket a typical speed-versus-reasoning trade-off faced in practice.

---

> > ### Author Response · Authors · 2025-08-01
> > **Update – Qwen3-32B TextGrad baseline now complete**
> >
> > Thank you for your patience while the Qwen3-32B TextGrad runs finished.
> > The results now can be inserted into above table:
> >
> > | Category | Method / Setting | LLM Model  | Swiss Acc | Swiss F1 | Swiss CE | Swiss AUC | Vaccine Acc | Vaccine F1 | Vaccine CE | Vaccine AUC |
> > |----------|-----------------|------------|-----------|----------|----------|-----------|-------------|------------|------------|-------------|
> > | **LLM-Based** | **TextGrad** | **Qwen3** | **0.5100** | **0.3669** | **2.0180** | **0.5322** | **0.4600** | **0.4276** | **6.4994** | **0.6303** |
> >
> > **Key take-away.**  After adding the TextGrad baseline of Qwen3-32B, ATHENA remains the top performer, reinforcing our main conclusion that coupling symbolic utility modeling with semantic adaptation yields the most accurate and interpretable results.

---

### Official Review · Reviewer_HnWo · 2025-07-02

**Clarity:** 3
**Significance:** 2
**Originality:** 4
**Rating:** 5
**Confidence:** 4

**Summary:**

This paper studies the problem of modeling human behaviour in complex domains where peoples idiosyncratic preferences may cause them to deviate from the global optima. As a solution to this problem the authors propose a two stage approach where first a global utility function is learned and then this function is tuned to each individuals unique characteristics. Both of these stages are implemented with the help of an LLM. They validate this approach on real world data showing this approach is performative and that both stages are important for success.

**Questions:**

In the experiment which tests the power of Individual-Level Semantic Adaptation alone, is it given a completely random starting point or the highest performing starting point out of a number of samples comparable to the amount that stage 1 generates? It seems unfair to attribute performance to step 1 that could be attributed to simply taking the best of N samples.

One of the primary goals of utility models, typically, is not to predict behaviour but to understand it. I see that in your fragment analysis some pieces of the predictive model are commonly reused but are any of the full models outputted by this procedure interpretable? Are the global functions interpretable?

Have you tried any more powerful models? In particular I would be interested in whether reasoning models show improved performance.

**Ethical Concerns:**

["NO or VERY MINOR ethics concerns only"]

**Final Justification:**

The authors rebuttal cleared up all of my concerns for the paper including comparing against stronger benchmarks and Ive therefore decided to bump my score to an accept.

**Limitations:**

yes

**Paper Formatting Concerns:**

no concerns

**Quality:**

3

**Strengths And Weaknesses:**

Strengths

- This is a very interesting and original idea
- The methodology seems sound
- The paper is well written and easy to follow
- The authors make a strong effort to ablate and interpret their results

Weaknesses

- The baselines compared against seem a little weak. Is there a reason why one couldnt train and end to end neural network?
- It would have been nice to see an ablation on the LLM component of the model. How much is the model benefiting from an LLM versus some other sampling procedure in the evolutionary algorithm and is it worth the additional computation cost

---

> ### Author Rebuttal · Authors · 2025-07-30
>
> We thank you for your time and effort in reviewing our paper! We find your suggestions very helpful and we address all your questions/comments as follows:
>
> > **W1/Q3: End-to-End Baseline Comparison & More Powerful Base Models**
>
> Thank you for the suggestions of adding end-to-end baselines and more powerful base models. We've been paying close attention to new cutting-edge models and have supplemented our experiments accordingly.
>
> - End-to-end models: BERT, TabNet, 3-layer-MLP.
> - Additional LLMs: Qwen3-32b, DeepSeek-R1-Distill-Qwen-32B, GPT-4o.
>
> Please see the updated results in the table published in rebuttal section of [Reviewer FBCn](https://openreview.net/forum?id=RDt0crdC7N&noteId=PIht69t6z8)
>
> > **Q2: Explainability of ATHENA utilities**
>
> We agree that interpretability is crucial and confirm that ATHENA yields fully interpretable, end-to-end utility functions. Below is two representative segments from the Swissmetro and Vaccine dataset.
>
> - **Representative Example — Swissmetro Dataset**
>
> | Mode  | Discovered symbolic utility|
> |-------|------------------------------------------|
> | **Train** | `K·(train_time + metro_time + luggage·log(age + 1) + age + is_male)  + C·(first_class + income)  − C·(GA_pass + headway)` |
> | **Car**   | `K·(car_time + train_time + luggage·log(age + 1) + age)             + C·(first_class + income)  − C·(GA_pass + metro_fare + is_male)` |
> | **Metro** | `K·(metro_time + luggage + age + is_male)                         + C·(first_class + income)  − C·(headway + GA_pass + is_male)` |
>
> Feature: Between 39 and 54 years old, identify as female, and have an income between 50 and 100.
>
> - **Key take-aways for domain experts**
>
> | Insight | Interpretation & policy relevance |
> |---------|-----------------------------------|
> | **Time dominates** | Large negative coefficients on travel-time variables show this segment is **highly time-sensitive** → investments that shorten door-to-door time (e.g., skip-stop service) should shift demand [1]. |
> | **Comfort premium** | Positive weight on `(first_class + income)` across all modes indicates a willingness to pay for comfort that scales with income → targeted upselling (seat reservations, quiet cars) is effective [2]. |
> | **Luggage burden grows with age** | The interaction `luggage·log(age + 1)` reveals baggage becomes disproportionately painful for older travelers → facilities such as luggage trolleys or porter services may raise train/metro share [3]. |
> | **GA pass effect** | Owning a GA pass (the Swiss national annual travel pass) biases travellers away from modes that still incur extra fares (e.g. Car, premium metro segments). Extending GA coverage to Swissmetro would therefore raise its relative appeal [4].|
>
> These insights translate raw coefficients into specific levers for service design and policy -- exactly the actionable value the reviewer is seeking. Taken together, these conclusions are highly valuable for practitioners and align closely with the extensive body of prior research on travel behavior and mode choice.
>
> [1] Shires, Jeremy D., and Gerard C. De Jong. "An international meta-analysis of values of travel time savings." Evaluation and program planning 32, no. 4 (2009): 315-325.
>
> [2] Abrantes, Pedro AL, and Mark R. Wardman. "Meta-analysis of UK values of travel time: An update." Transportation Research Part A: Policy and Practice 45, no. 1 (2011): 1-17.
>
> [3] Chang, Yu-Chun. "Factors affecting airport access mode choice for elderly air passengers." Transportation research part E: logistics and transportation review 57 (2013): 105-112.
>
> [4] Weis, Claude, Kay W. Axhausen, Robert Schlich, and René Zbinden. "Models of mode choice and mobility tool ownership beyond 2008 fuel prices." Transportation Research Record 2157, no. 1 (2010): 86-94.
>
> - **Representative Example — Vaccine Dataset**
>
> | Mode  | Discovered symbolic utility|
> |-------|------------------------------------------|
> | **Unvaccinated** | `C·covid_threat·(1 + trust_government·trust_science·log(age + 5))·risk_of_covid_greater_than_vax  +  K·have_covid_sick_family_member·log(age + 4)` |
> | **Vaccinated (no booster)**   | `C·covid_threat + C·vaccine_safe_to_me  +  K·(trust_government·trust_science·more_attention_to_vax_info·√(age + 4))` |
> | **Booster** | `C·e^{age^1.5}·covid_threat·√(vax_protect_long_yes)  +  C·vaccine_safe_to_me  +  K·(trust_government·trust_science·nurse·√(age + 7))` |
>
> Feature: Age 18–38, income above county median.
>
> - **Key take-aways for domain experts**
>
> | Insight | Interpretation & policy relevance |
> |---------|-----------------------------------|
> | **Risk trade-off in vaccination choice** | The product `covid_threat × risk_of_covid_greater_than_vax` captures a critical decision-making trade-off: individuals weigh the perceived risk of infection against their belief about vaccine safety. Effective messaging must work to narrow this perceived risk gap, for instance by emphasizing robust evidence on vaccine safety and the serious consequences of infection [5]. |
> | **Booster demand rises steeply with age** | The factor $e^{age^{1.5}}$ generates a nonlinear age effect: as age increases, the perceived benefit of taking the vaccine grows rapidly. This pattern likely reflects age-associated increases in risk perception and underlying health vulnerabilities [6]. |
> | **Prior belief and healthcare occupation** | The presence of `vax_protect_long_yes` and nurse occupation in the booster equation means emphasizing extended protection and occupation will push this group further along the vaccination ladder [7]. |
> | **Trust is pivotal for vaccine uptake** | The multiplicative `trust_government × trust_science` term appears in every vaccinated utility, signalling that confidence in both institutions—not just one—amplifies willingness [8].|
>
> These findings convert raw symbolic coefficients into tailored interventions: age-targeted messaging, trust-building coalitions, and occupation-based booster incentives.
>
> [5] Green, Manfred S. "Rational and irrational vaccine hesitancy." Israel Journal of Health Policy Research 12, no. 1 (2023): 11.
>
> [6] Noh, Yunha, Ju Hwan Kim, Dongwon Yoon, Young June Choe, Seung-Ah Choe, Jaehun Jung, Sang-Won Lee, and Ju-Young Shin. "Predictors of COVID-19 booster vaccine hesitancy among fully vaccinated adults in Korea: a nationwide cross-sectional survey." Epidemiology and Health 44 (2022): e2022061.
>
> [7] Biswas, Nirbachita, Toheeb Mustapha, Jagdish Khubchandani, and James H. Price. "The nature and extent of COVID-19 vaccination hesitancy in healthcare workers." Journal of community health 46, no. 6 (2021): 1244-1251.
>
> [8] Trent, Mallory, et al. "Trust in government, intention to vaccinate and COVID-19 vaccine hesitancy: A comparative survey of five large cities in the United States, United Kingdom, and Australia." Vaccine 40.17 (2022): 2498-2505.
>
>
> > **W2: Ablation on the LLM component**
>
> Thank you for the suggestion regarding an ablation of the LLM component. We appreciate the opportunity to clarify the role and necessity of the LLM in our framework. 1) Our LLM-based utility search relies on two distinct libraries: a Symbolic Library, which contains standard algebraic operators, and a Concept Library, which includes semantically rich, high-level terms derived from domain-specific text. The Concept Library is only accessible through the LLM. 2) Ablating the LLM would not simply remove a component of the model; it would eliminate access to the Concept Library entirely. This makes a clean ablation infeasible: any such comparison would conflate the removal of the LLM with the removal of high-level semantic concepts, thereby blurring the effect under study. 3) Traditional symbolic regression methods (e.g., GP, SRBench) rely solely on the Symbolic Library and therefore cannot propose or reason over high-level, text-derived concepts. As a result, their outputs often lack the semantic richness and real-world relevance that LLM-enhanced models provide. This is not an apples-to-apples comparison, and such a comparison would overlook the core novelty of our approach.
>
> In short, the LLM is not just a computational enhancement; it is essential to the inclusion of semantically meaningful, multimodal domain knowledge in the search process. We hope this clarifies the role of the LLM and why a traditional ablation would not offer a fair ablation. That said, we are open to alternative suggestions for more targeted comparisons, should the reviewer have specific ideas in mind.
>
> > **Q1: Initialization in the *Stage-2-only* Ablation**
>
> **Paper setting.**
> For the “Individual-Level Semantic Adaptation only” ablation we seeded TextGrad with **one** symbolic template drawn **uniformly at random**. No *best-of-N* filtering was applied.
>
> - **New control experiment: *best / median / worst* of all formulas after 30 iterations**
>
> We sampled 10 individuals from Swissmetro datasets. We selected the best, median, and worst symbolic utility formula after 30 iterations of Stage 1. We then ran TextGrad on these formulas as initial seeds.
>
> | LLM                | Init‑seed | Swissmetro Acc. (mean ± std) | Swissmetro F1 (mean ± std) |
> |--------------------|-----------|---------------------------|--------------------------|
> | **GPT‑4o‑mini**    | best   | 0.8611 ± 0.1273 | 0.4127 ± 0.1803 |
> |                    | median | 0.8333 ± 0.2887 | 0.4074 ± 0.2313 |
> |                    | worst  | 0.6250 ± 0.2320 | 0.3558 ± 0.2319 |
>
> **Key Observation** After TextGrad refinement, median-quality seeds converge to within ≈ 3 pp Acc and ≈ 0.6 pp F1 of the best seed, while even the worst seed retains ≈ 73 % of the best accuracy. Hence Stage 2’s boost comes from gradient adaptation, not from simply picking the “best of N” start.

---

> > ### Comment · Reviewer_HnWo · 2025-08-01
> >
> > This has cleared up the majority of my concerns. Thank you for the response!

---

> > > ### Author Response · Authors · 2025-08-01
> > >
> > > We’re glad our rebuttal resolved most of your concerns. If any additional questions arise, we’re happy to clarify further. We really appreciate your thorough review and constructive feedback!

---

### Official Review · Reviewer_F7TL · 2025-07-03

**Clarity:** 4
**Significance:** 3
**Originality:** 3
**Rating:** 5
**Confidence:** 3

**Summary:**

This paper proposes ATHENA (Adaptive Textual-symbolic Human-centric Reasoning), a framework for personalized decision modeling that combines symbolic utility discovery and LLM-powered semantic adaptation. The approach first uses LLM-augmented symbolic regression to derive interpretable group-level utility functions for predefined demographic groups, then refines these models at the individual level via TextGrad to capture personal preferences and constraints. Evaluations on transportation mode choice and COVID-19 vaccine uptake tasks show that ATHENA outperforms classical utility-based models, machine learning classifiers, and LLM baselines by at least 6.5% in F1 score. The method is notable for producing interpretable, human-centric models, but its reliance on manually specified demographic groupings limits its flexibility and raises questions about the robustness of these partitions across diverse domains.

**Questions:**

- Are $T$ and $T’$ two hyper-parameters? What are the chosen hyper-parameters for the experiments? For Figure 3, is the iteration (x-axis) showing $T$ or $T’$?

## Suggestions

- The “Stage 1” and “Stage 2” annotations in Algorithm 1 currently look like part of the code; consider changing their style (e.g., italics or actual comment syntax) so they visually appear as comments and are easier to distinguish.
- Style-wise, the `enumerate` on line 133 could have **shorter left padding** to improve visual alignment and readability.
- The inline citations on lines 235–238 are attached directly to the previous word; there should be **one space separating the citations** from the preceding text for proper formatting.
- Include at least **one qualitative example in the main text** showcasing a discovered group-level symbolic utility formula, along with **proper takeaways or insights**. This is crucial to demonstrate the value of having an explainable model for informing domain experts (e.g., public health decision-makers).
- Provide more context on why **Cross Entropy (CE)** is included as a metric for evaluating ATHENA. Given its meaning in probabilistic models, explain its relevance to your setting and why it complements metrics like F1 and AUC.

**Ethical Concerns:**

["NO or VERY MINOR ethics concerns only"]

**Final Justification:**

This paper proposes ATHENA (Adaptive Textual-symbolic Human-centric Reasoning), a framework for personalized decision modeling that combines symbolic utility discovery and LLM-powered semantic adaptation. During review, I raised questions about the reliance on demographic groups and the derived symbolic representations. Through rebuttal, the authors have justified that such setting is reasonable, and provided qualitative analysis on symbolic representations. I'm happy that my concerns are resolved and that such changes are going to be made in the final version of the paper. I'm raising my rating of the paper (4 -> 5).

**Limitations:**

- Computational Complexity: Textual gradient optimization is resource-intensive, limiting scalability to large populations.
- Predefined Group Assignment: Requires a priori demographic groups, which may oversimplify heterogeneity and miss latent patterns.
- Group Significance: Lacks a principled method to verify whether demographic partitions are behaviorally meaningful, risking weak or biased utility functions.

**Quality:**

3

**Strengths And Weaknesses:**

## Strengths

- **The paper is very well written** and easy to follow, with clear motivation, technical details, and experimental narrative.
- **Visualizations are excellent**, effectively illustrating the two-stage ATHENA pipeline and the learned symbolic components.
- The proposed framework combines symbolic utility modeling with LLM-driven semantic adaptation in a **conceptually elegant and technically sound way**.
- **Premise is strong**: addressing the gap between population-level utility models and personalized decision-making is both timely and impactful.
- **Performance is very good**, achieving ≥6.5% F1 score improvement over classical discrete choice models, machine learning baselines, and LLM-only approaches on real-world datasets.
- The two-stage design (group-level symbolic utility discovery and individual-level semantic adaptation) is **well-motivated and validated** through ablation studies, demonstrating clear complementarity.
- Produces **interpretable, human-centric decision models** with potential relevance for high-stakes applications such as public health policy and transportation planning.

## Weaknesses

- The framework relies on predefined demographic groups for the group-level symbolic utility discovery stage, which assumes these partitions are optimal and behaviorally meaningful. This is akin to providing the top-level splits in a decision tree without verifying their significance, limiting flexibility and potentially propagating bias.
    - Is the baselines also provided the same grouping during the experiments?
- There is no mechanism to automatically discover or validate significant groupings, which raises concerns about robustness in domains with high intra-group heterogeneity.
- While the integration of symbolic regression and LLMs is novel, the individual components (symbolic regression, TextGrad) are adaptations of existing methods rather than fundamentally new techniques.
- The paper could benefit from more qualitative analysis and takeaways from the synthesized group-level utility formulas. Since one of the main advantages of having an interpretable model is to derive actionable insights for domain experts (e.g., public health professionals), this feels like a missed opportunity to showcase the framework’s real-world utility beyond predictive performance.
- (Minor) The paper has not discussed scalability of the method. When $T$ and $T'$ are treated as fixed hyper-parameters, and disregarding LLM inference time, is the methodology a constant time algorithm? Either a high level discussion or an empirical evaluation (including LLM invocation time) would be appreciated.

I am happy to increase scores when the questions are well-addressed.

---

> ### Author Rebuttal · Authors · 2025-07-30
>
> We thank you for your time and effort in reviewing our paper! We find your suggestions very helpful and we hereby address your questions:
>
> > **W1/W2 & L2/L3 – Predefined Demographic Grouping & Robustness**
>
> To clarify, all baseline models received the same demographic features and groups. Our choice to use these groups (common demo-based group) was a deliberate decision motivated by:
>
> - **Theoretical Grounding**: It aligns with established practices in decision choice modeling researches that use demographics to explain behavioral heterogeneity [1-3].
>
> - **Interpretability**: It yields human-readable symbolic rules that translate directly into levers. E.g., for women aged 39-54 earning 50–100 k CHF, the rules reveal that shorter travel-time, premium comfort, and baggage assistance most sway their choice -- actionable insight that guides operators toward faster services, targeted upsells, and better luggage support for this segment.
>
> - **Robustness**: Automatically discovering behaviorally consistent latent clusters is a challenging task that typically requires large datasets. Given the scale of our experimental data, relying on established demographic priors provides a more robust initial partition. It mitigates the risk of overfitting and learning spurious patterns. Also, we don't want to increase complexity or introduce additional hyperparameters to make the model more difficult to use.
>
> Crucially, our ablation study shows that removing the group-level symbolic discovery stage causes a severe performance drop. This result underscores that a well-structured, group-informed prior is not a limitation but a necessary component for the model's success, preventing the unguided adaptation from converging to poor local optima.
>
> While we agree automated group discovery is a valuable direction for future research, our work demonstrates the marked effectiveness of this principled, two-stage approach.
>
> [1] Tymula, Agnieszka, et al. "Adolescents’ risk-taking behavior is driven by tolerance to ambiguity." Proceedings of the National Academy of Sciences 109.42 (2012): 17135-17140.
>
> [2] De Bruijn, Ernst-Jan, and Gerrit Antonides. "Poverty and economic decision making: a review of scarcity theory." Theory and Decision 92, no. 1 (2022): 5-37.
>
> [3] Croson, Rachel, and Uri Gneezy. "Gender differences in preferences." Journal of Economic literature 47, no. 2 (2009): 448-474.
>
> > **W3: Component novelty**
>
> The contribution should be evaluated on the end-to-end paradigm and its conceptual novelty, not on isolated components. As noted in Section 1, population-level models often overlook each person’s unique cognitive calculus -- the belief, the unique constraints and the context-specific trade-offs that drive real decisions. ATHENA captures this calculus through a novel two-stage pipeline: symbolic regression extracts compact, group-level utility laws, and TextGrad then personalizes them to each individual.
> - Symbolic regression for insight: Generates interpretable, domain-aware formulas that reveal feature relations far beyond merely black-box predictors.
> - Informed semantic optimization: Uses those formulas as data-driven priors, mitigating TextGrad’s sensitivity to initialization and yielding faster, more reliable convergence to personalized models.
>
> > **W4: Qualitative Analysis of symbolic representations**
>
> We agree that deeper qualitative analysis and actionable takeaways from our group-level utility formulas would strengthen the manuscript. Below, we provide a detailed qualitative analysis of the example Swissmetro symbolic utilities. For a parallel example on the Vaccine dataset, see the “Q2: Explainability of ATHENA utilities” section in [Reviewer HnWo’s rebuttal](https://openreview.net/forum?id=RDt0crdC7N&noteId=STQUOcL5iO). These examples demonstrate that our group-level utilities are both interpretable and directly actionable for domain experts.
>
> - **Representative Example — Swissmetro Dataset**
>
> | Mode  | Discovered symbolic utility|
> |---|---|
> | **Train** | `K·(train_time + metro_time + luggage·log(age + 1) + age + is_male)  + C·(first_class + income)  − C·(GA_pass + headway)` |
> | **Car**   | `K·(car_time + train_time + luggage·log(age + 1) + age)             + C·(first_class + income)  − C·(GA_pass + metro_fare + is_male)` |
> | **Metro** | `K·(metro_time + luggage + age + is_male)                         + C·(first_class + income)  − C·(headway + GA_pass + is_male)` |
>
> Feature: Between 39 and 54 years old, identify as female, and have an income between 50 and 100.
>
> - **Key take-aways for domain experts**
>
> | Insight | Interpretation & policy relevance |
> |---|---|
> | **Time dominates** | Large negative coefficients on travel-time variables show this segment is **highly time-sensitive** → investments that shorten door-to-door time (e.g., skip-stop service) should shift demand [1]. |
> | **Comfort premium** | Positive weight on `(first_class + income)` across all modes indicates a willingness to pay for comfort that scales with income → targeted upselling (seat reservations, quiet cars) is effective [2]. |
> | **Luggage burden grows with age** | The interaction `luggage·log(age + 1)` reveals baggage becomes disproportionately painful for older travelers → facilities such as luggage trolleys or porter services may raise train/metro share [3]. |
> | **GA pass effect** | Owning a GA pass (the Swiss national annual travel pass) biases travellers away from modes that still incur extra fares (e.g. Car, premium metro segments). Extending GA coverage to Swissmetro would therefore raise its relative appeal [4].|
>
> These insights translate raw coefficients into specific levers for service design and policy -- exactly the actionable value the reviewer is seeking. Taken together, these conclusions are highly valuable for practitioners and align closely with the extensive body of prior research on travel behavior and mode choice.
>
> [1] Shires, Jeremy D., and Gerard C. De Jong. "An international meta-analysis of values of travel time savings." Evaluation and program planning 32, no. 4 (2009): 315-325.
>
> [2] Abrantes, Pedro AL, and Mark R. Wardman. "Meta-analysis of UK values of travel time: An update." Transportation Research Part A: Policy and Practice 45, no. 1 (2011): 1-17.
>
> [3] Chang, Yu-Chun. "Factors affecting airport access mode choice for elderly air passengers." Transportation research part E: logistics and transportation review 57 (2013): 105-112.
>
> [4] Weis, Claude, Kay W. Axhausen, Robert Schlich, and René Zbinden. "Models of mode choice and mobility tool ownership beyond 2008 fuel prices." Transportation Research Record 2157, no. 1 (2010): 86-94.
>
> > **Q1: Hyper-parameter clarification**
>
> Yes. In ATHENA both $T$ and $T’$ are user-set hyper-parameters:
> - $T$ – maximum iterations of Stage 1: Group-level Symbolic Utility Discovery.
> - $T’$ – maximum iterations of Stage 2: Individual-level Semantic Adaptation.
>
> Figure 3 tracks the symbolic-discovery loop, so its x-axis is $T$ (30 iterations in our runs).
>
> | Symbol | Role in ATHENA | Value used |
> | :---:  | --- | :--- |
> | LLMs | Backbone models (both stages) | `gpt-4o-mini`, `gemini-2.0-flash` |
> | $T$ | Iterations in Stage 1 (Fig. 3) | **30** |
> | $T'$ | Iterations in Stage 2 | **5** |
> | $K$ | Candidate utilities per Stage 1 iteration | **10** |
>
> > **W5 & L1: Scalability and parameter $T$ and $T'$**
>
> #### 1. Theoretical analysis
>
> With $T$, $T'$ fixed, the algorithm is linear, not constant:
>
> $$
> t_{\text{total}}
> = t_{\text{Stage1}} + t_{\text{Stage2}}
> \sim
> \underbrace{\lvert\mathcal{G}\rvert\cdot K\cdot T}\_{\text{symbolic utility search}}
> +
> \underbrace{N\cdot T'}\_{\text{textual gradient refinement}}
> $$
>
> Each term is multiplied by the average LLM latency $\tau$ (roughly linear in token count):
>
> $$
> \mathcal{O}\bigl((\lvert\mathcal{G}\rvert K T + N T')\,\tau_{\text{tok}}\bigr)\;.
> $$
>
> Both stages can be parallelized:
> * **Stage 1:** symbolic searches for each demographic group run independently.
> * **Stage 2:** TextGrad refinements for each individual can be batched or dispatched to separate workers.
>
> #### 2. Empirical evidence
> Using **gpt-4o-mini**, we measured wall-clock time and token usage on the Swissmetro subset:
>
> Stage 2 – Individual adaptation
>
> | $T'$ (iterations) | Stage-2 time (s) | s / iter | tokens / iter |
> |:---:|:---:|:---:|:---:|
> |1|48.16|48.16 | 1079.6 |
> |3|176.89|58.96| 1195.83 |
> |5|315.66|63.13| 1249.28 |
>
> Stage 1 – Group-level discovery
>
> |$T$ (iterations)|Stage-1 time (min)|tokens total|
> |:---:|:---:|:---:|
> |5|30.61|215281.3|
> |15|36.69|251974.1|
> |30|65.64|479751.3|
>
> Here we can see runtime and tokens grow linearly with iterations: Stage 2: 48 s → 316 s (T′ 1 → 5); Stage 1: 31 min → 66 min (T 5 → 30)—so a full Swissmetro run (T = 30, T′ = 5) finishes in < 70 min and ≈ 0.5 M tokens on one worker, with near-perfect parallel speed-up.
>
> > **S1\S2\S3\S4: Paper Writing**
>
> Thank you for the helpful suggestions. We have revised all relevant instances in the manuscript. The updates are summarized below.
>
> |Position|Original | Revised|
> | :--- | :---| :---|
> |Algorithm 1, Near L198|**Stage 1:** …|*// Stage 1 – Group-Level Symbolic Discovery*|
> |Algorithm 1, Near L208|**Stage 2:** …|*// Stage 2 – Individual-Level Semantic Adaptation*|
> |Page 3, L133, L137| ` 1. …  2. ...`   | `1. … 2. ...` (reduced left padding)|
> |Page 7, L235–238| eg. "…model [98,99] and…", "...few-shot method[100,101]"...| "…model [98, 99] and …" (added spacing before citations), "...few-shot method [100, 101]"... |
>
> > **S5: Cross Entropy metric**
>
> Cross-Entropy is important because the ATHENA models decision-making as a probabilistic choice, consistent with the Random Utility Maximization theory. A lower CE value shows the model assigns higher probabilities to the choices individuals actually make.
>
> While F1/AUC measures classification performance, CE provides a distinct and complementary evaluation of the model's confidence calibration. With all those metrics, we can better understand the model's performance and robustness.

---

> > ### Comment · Reviewer_F7TL · 2025-08-05
> >
> > Thank you for your detailed and thoughtful response. I am satisfied with the answers and please make sure to add these information into your final version of the paper. I will keep my positive score.

---

> > > ### Author Response · Authors · 2025-08-06
> > >
> > > Dear Reviewer F7TL,
> > >
> > > We sincerely appreciate your detailed and thoughtful feedback. Your suggestions have significantly contributed to the clarity and depth of our work!
> > >
> > > We are very glad to hear that our responses addressed your concerns and that you are satisfied with the current revision. Of course! We will ensure that all the discussed improvements—including hyperparameter analysis, qualitative analysis of symbolic representations, and formatting refinements—are carefully incorporated into the final version.
> > >
> > > Thank you again for your constructive comments! Please feel free to reach out if you have any further thoughts or suggestions—we would be more than happy to engage in further discussion.
> > >
> > > Sincerely,
> > > The Authors

---

### Note · Authors · 2025-08-12

We thank all reviewers and the AC for thoughtful feedback that strengthened our work. Below, we summarize the final paper, clarifications, and contributions.

## 1. Core Contribution
ATHENA models personalized decision-making in two stages:
1. **Group-Level Symbolic Utility Discovery** – Uses LLM-augmented symbolic discovery to find robust, interpretable utility functions for demographic groups.
2. **Individual-Level Semantic Adaptation** – Integrate optimal group-level utility functions with unstructured preferences and constraints using personalized semantic templates to enable individualized decision-making.

On real-world travel mode and vaccine choice tasks, ATHENA achieves **≥6.5% F1** gain over the strongest utility-based, ML, and LLM baselines, yielding models that are both predictive and interpretable for targeted interventions.

## 2. Addressed Concerns
- **Predefined groups & robustness**: All baselines were provided with the same demographic features. We adopted established demographic partitions, which are widely used in discrete choice modeling, to ensure behavioral interpretability and mitigate overfitting risks on limited datasets.
- **Model scope**: Added three end-to-end baselines (BERT, TabNet, MLP) and three additional LLMs (Qwen3-32B, DeepSeek-R1-Distill-Qwen-32B, GPT-4o). ATHENA remains top across accuracy, F1 and AUC.
- **Scalability & cost**: Reported runtime/token costs, including budget for automatic retries; costs are modest with near-linear scaling and high parallelism.
- **Interpretability**: Added multiple utility formulas with domain takeaways for public health intervention and transportation planning, showing interpretability beyond predictive metrics.
- **Hyperparameters**: All key settings and iteration counts are now reported.

## 3. Broader Impact
ATHENA bridges population-optimal decisions and individual choices by combining symbolic utility modeling with semantic adaptation in LLM, advancing equity, transparency, and transferability in decision support. This allows decision-makers to:
- Support actionable, equity-aware decisions (e.g., tailor vaccine outreach by trust profile; optimize transport for time-sensitive groups).
- Interpret decisions via explicit, domain-relevant formulas.
- Generalize to different domains where structured and unstructured information jointly shape decision outcomes (public health, transportation, disaster response).

---

### Decision · Program_Chairs · 2025-09-17

**Decision:**

Accept (spotlight)

**Comment:**

The paper tackles the problem of bridging the gap between population-level utility models and individual decision-making through a two-stage framework combining symbolic utility discovery with LLM-powered semantic adaptation (termed ATHENA).

In summary, the reviews suggest that this work hits a sweet spot of methodological innovation and practical relevance. The authors elegantly sidestep the false dichotomy between interpretable-but-rigid utility models and flexible-but-opaque neural approaches. Reviewers specifically comment on several strengths. Among others:
- The quality of writing and presentation
- Strong premise/motivation
- Good performance
- Interpretability
- Originality of the idea

The authors also made a commendable effort in addressing reviewer concerns by adding stronger baselines and cost analysis, the latter of which shows the approach is practically viable even under budget constraints.

There were a few weaknesses noted:
- The reliance on predefined demographic groups is both a strength and limitation. Though this is well-justified theoretically, it does constrain generalizability
- More importantly, the evaluation is limited to two domains with relatively small datasets. The claim of broad applicability would ideally be backed by stronger empirical support across diverse contexts
- The symbolic discovery stage, while novel in its LLM integration, essentially performs a kind of guided search over a constrained space. The innovation lies more in the orchestration than a fundamental algorithmic advance

Overall, my view is that this paper advances an important research direction with a technically sound approach and clear promise for high-stakes applications. The authors have also responded well to many of the raised concerns. I think this is a clear accept.